# MutSα maintains the mismatch repair capability by inhibiting PCNA unloading

Yoshitaka Kawasoe[1], Toshiki Tsurimoto[2], Takuro Nakagawa[1], Hisao Masukata[1], Tatsuro S Takahashi[1]*

[1]Graduate School of Science, Osaka University, Toyonaka, Japan; [2]Department of Biology, Faculty of Sciences, Kyushu University, Fukuoka, Japan

**Abstract** Eukaryotic mismatch repair (MMR) utilizes single-strand breaks as signals to target the strand to be repaired. DNA-bound PCNA is also presumed to direct MMR. The MMR capability must be limited to a post-replicative temporal window during which the signals are available. However, both identity of the signal(s) involved in the retention of this temporal window and the mechanism that maintains the MMR capability after DNA synthesis remain unclear. Using *Xenopus* egg extracts, we discovered a mechanism that ensures long-term retention of the MMR capability. We show that DNA-bound PCNA induces strand-specific MMR in the absence of strand discontinuities. Strikingly, MutSα inhibited PCNA unloading through its PCNA-interacting motif, thereby extending significantly the temporal window permissive to strand-specific MMR. Our data identify DNA-bound PCNA as the signal that enables strand discrimination after the disappearance of strand discontinuities, and uncover a novel role of MutSα in the retention of the post-replicative MMR capability.

*For correspondence: tatsuro_takahashi@bio.sci.osaka-u. ac.jp

Competing interests: The authors declare that no competing interests exist.

## Introduction

The evolutionarily conserved mismatch repair (MMR) system corrects replication errors post-replicatively to prevent their being fixed as mutations in the next round of complementary DNA synthesis (*Iyer et al., 2006*; *Jiricny, 2013*; *Kunkel and Erie, 2015*). Since erroneously inserted nucleotides are present on newly synthesized DNA strands, identification of the newly synthesized strand is a critical step in MMR. By rectifying replication errors, MMR increases the replication fidelity by ~2 orders of magnitude in yeast (*Lujan et al., 2014*). In humans, genetic or epigenetic inactivation of MMR genes elevates the risk of tumorigenesis in both hereditary and sporadic manners (*Jiricny, 2013*).

In *Escherichia coli*, MMR distinguishes the parental and daughter strands mainly by using adenine methylation on GATC sites (*Lahue et al., 1989*; *Iyer et al., 2006*). Mismatches are recognized by the mismatch sensor MutS homodimer. MutS and the MutL homodimer then activate the latent nicking-endonuclease MutH, which cleaves the unmethylated strand at the hemi-methylated GATC sequences. MMR is possible from the time of hemi-methylated GATC generation by DNA synthesis until full methylation of the GATC sites. Maintaining this temporal window is critical for efficient MMR, because over-expression of the Dam methylase, by which full-methylation of the GATC sites is accelerated, significantly elevates the mutation frequency (*Herman and Modrich, 1981*; *Marinus et al., 1984*). The *E. coli* MMR system can also correct replication errors through a methylation-independent mechanism, where strand discontinuities can substitute for GATC methylation both in vivo and in vitro (*Laengle-Rouault et al., 1986*; *Lahue et al., 1987*; *1989*).

Eukaryotic MMR is directed by strand discontinuities such as nicks or gaps in vitro (*Holmes et al., 1990*; *Thomas et al., 1991*). Two MutS heterodimers, MutSα (Msh2-Msh6) and MutSβ (Msh2-Msh3) recognize replication errors; MutSα has a biased preference for base-base mismatches and small insertion/deletion loops (IDLs), while MutSβ preferentially recognizes large IDLs (*Iyer et al., 2006*;

**eLife digest** To pass on genetic information from one generation to the next, the DNA in a cell must be precisely copied. DNA is made of two strands and genetic information is encoded by sequences of molecules called bases in the strands. The bases from one strand form pairs with complementary bases on the other strand. However, errors in the copying process result in unmatched pairs of bases. Such errors are corrected by a repair system called mismatch repair.

When DNA is copied, the two strands are separated and used as templates to make new complementary strands. This means that errors only arise on the new strands. Mismatch repair must therefore target the new strands to maintain the original information encoded by the template DNA. The repair needs to happen before the copying process is complete because the template strands and the new strands become indistinguishable afterwards. However, it is not clear how the two processes communicate with each other.

Previous studies have identified a ring-shaped molecule called the replication clamp – which is essential for the copying process – as a prime candidate for the molecule responsible for this communication. This molecule binds to the DNA to promote the copying process, and afterwards it is removed from the DNA by other molecules. Furthermore, a group of proteins called the MutSα complex, which recognizes unmatched bases in DNA molecules, physically interacts with the replication clamp.

Kawasoe et al. used eggs from African clawed frogs to study how the replication clamp connects the copying process and mismatch repair in more detail. The experiments show that when the replication clamp is bound to the DNA, it is able to direct mismatch repair to a specific DNA strand. When MutSα recognizes unmatched bases, it prevents the replication clamp from being removed from the DNA. By doing so, MutSα prevents the information about the new DNA strand from being lost until mismatch repair has taken place.

These findings reveal new interactions between DNA copying and the correction of errors by mismatch repair. The next steps will be to understand how MutSα is able to keep the replication clamp on the DNA and to clarify its role in protecting DNA from gaining mutations.

*Jiricny, 2013*; *Kunkel and Erie, 2015*). After mismatch binding, MutSα/β are converted into closed 'clamp-like' forms, by which they can translocate along DNA. They then recruit the latent nicking-endonuclease MutLα (Mlh1-Pms2 in vertebrates and Mlh1-Pms1 in yeast) to initiate the removal of the error-carrying DNA strand. Two other eukaryotic MutL homologs, MutLβ (Mlh1-Pms1 in vertebrates and Mlh1-Mlh2 in yeast) and MutLγ (Mlh1-Mlh3) are suggested to play minor roles in somatic MMR (*Jiricny, 2013*; *Campbell et al., 2014*). Successful reconstitutions of eukaryotic MMR have shown that MutSα, or MutSβ, and MutLα, the Proliferating Cell Nuclear Antigen (PCNA) sliding clamp, the clamp loader Replication Factor C (RFC), the Exo1 exonuclease, and the DNA synthesis components promote MMR when a strand discontinuity is present (*Genschel and Modrich, 2003*; *Dzantiev et al., 2004*; *Constantin et al., 2005*; *Zhang et al., 2005*). A strand discontinuity, which can occur on either 5'- or 3'-side of the mismatch, activates MutLα through a MutSα- and PCNA-dependent mechanism to induce successive rounds of nicking on the error-carrying strand (*Kadyrov et al., 2006*; *2007*; *Pluciennik et al., 2010*). Single-strand DNA termini such as 5'-ends of the Okazaki fragments serve as entry points for Exo1 and strand discrimination signals in vivo as well (*Pavlov et al., 2003*; *Nick McElhinny et al., 2010*; *Liberti et al., 2013*; *Lujan et al., 2014*; *Liu et al., 2015*). Recent studies have revealed that ribonucleotides embedded by replicative DNA polymerases serve as strand signals for MMR in vitro and contribute to a sub-fraction of leading strand MMR in vivo, after converted into single-strand gaps through RNaseH2-dependent ribonucleotide excision repair (*Ghodgaonkar et al., 2013*; *Lujan et al., 2013*).

PCNA has also been presumed to be the strand discrimination marker in eukaryotes (*Umar et al., 1996*; *Chen et al., 1999*; *Pavlov et al., 2003*; *Dzantiev et al., 2004*; *Kadyrov et al., 2006*; *Pluciennik et al., 2010*; *Hombauer et al., 2011b*; *Peña-Diaz and Jiricny, 2012*; *Georgescu et al., 2015*; *Kunkel and Erie, 2015*). PCNA is a ring-shaped homo-trimer that supports various DNA transactions including DNA replication and repair (*Georgescu et al., 2015*). PCNA is loaded onto DNA

from the template-primer junction by RFC, and likely unloaded by an RFC-like complex containing Elg1 after the completion of DNA synthesis (*Kubota et al., 2013*; *2015*). Since its DNA binding is asymmetric with respect to polarities of the parental and daughter strands, DNA-bound PCNA can hold information for the newly synthesized strand (*Bowman et al., 2004*; *Georgescu et al., 2015*). PCNA plays an essential role in an early MMR step that precedes degradation of the error-carrying strand (*Umar et al., 1996*). PCNA loaded from a strand discontinuity induces strand-specific, mismatch- and MutSα-dependent activation of the MutLα endonuclease (*Kadyrov et al., 2006*; *2007*; *Pluciennik et al., 2010*). When PCNA is loaded onto closed circular DNA without defined orientation, it induces unbiased nicking on both DNA strands (*Pluciennik et al., 2010*; *2013*). These findings have led to a proposal that orientation of DNA-bound PCNA is a critical determinant for the nicking specificity of MutLα.

In addition to its proposed role in strand discrimination, PCNA is also involved in multiple steps of MMR. Numbers of PCNA mutants in yeast exhibit mutator phenotypes that are epistatic to MMR mutations (*Johnson et al., 1996*; *Umar et al., 1996*; *Chen et al., 1999*; *Amin et al., 2001*; *Lau et al., 2002*; *Goellner et al., 2014*). It interacts with MutSα/β through their PCNA-interacting peptide (PIP) motifs, which reside at the N-termini of Msh6 and Msh3 (*Clark et al., 2000*; *Flores-Rozas et al., 2000*; *Kleczkowska et al., 2001*). In both cases, the PIP motifs and the mispair binding domains are connected through long linkers, which are predicted to be disordered in yeast (*Shell et al., 2007*). Mutants of the PIP motifs in Msh6 and Msh3 show substantial but not complete reduction of the MMR activity, indicating that the PIP motif plays an important, yet assistive, non-essential role in MMR (*Clark et al., 2000*; *Flores-Rozas et al., 2000*). The PIP motif has been proposed to tether MutSα to the replication fork to assist its mismatch recognition (*Kleczkowska et al., 2001*; *Hombauer et al., 2011a*; *Haye and Gammie, 2015*). A recent study has shown that this motif in MutSα is important for a late MMR step(s) involving degradation of the error-carrying strand, especially by the Exo1-independent mechanism (*Goellner et al., 2014*). These findings suggest that PCNA coordinates multiple reactions in MMR, yet the exact role of PCNA in MMR is still not clearly defined.

An important remaining question in eukaryotic MMR is how (and how long) the strand discrimination signals are retained after DNA synthesis. It has been shown in yeast that MutSα must be present during S-phase to suppress mutations (*Hombauer et al., 2011b*). Therefore, the generation and retention of the strand signals for MMR should be intimately coupled with DNA replication. Since PCNA transiently remains on DNA after the completion of DNA synthesis, DNA-bound PCNA could mediate the coupling of MMR with DNA replication. However, because experimental evidence that DNA-bound PCNA induces strand-specific MMR in the absence of strand discontinuities is currently lacking, it is still uncertain whether PCNA by itself can mediate the coupling of MMR with DNA replication. In addition, how functional interaction between the MMR system and DNA-bound PCNA, which should be unloaded from DNA shortly after the completion of DNA synthesis, is ensured remains highly unclear.

Here, we demonstrate that MutSα and PCNA play a critical role in maintenance of the MMR capability after DNA synthesis. We show that nucleoplasmic extract (NPE) of *Xenopus* eggs efficiently induces gap-directed, strand-specific MMR, whose capability remains even after sealing of the gap. We further show that DNA-bound PCNA induces strand-specific MMR in the absence of strand discontinuities. Strikingly, we found that MutSα attenuates unloading of PCNA and retains the MMR capability largely through its PIP motif. Our data thus identify PCNA as the eukaryotic strand discrimination molecule that retains the MMR capability after DNA synthesis, and delineate a critical role of MutSα in maintenance of the temporal window for eukaryotic MMR.

## Results

### Nucleoplasmic extract of *Xenopus* eggs efficiently promotes both gap-directed MMR and gap-filling DNA synthesis

The nucleoplasmic extract (NPE), a highly concentrated nuclear protein extract of *Xenopus* eggs, has been used as a powerful in vitro model system for DNA repair reactions coupled with DNA synthesis (*Walter et al., 1998*; *Räschle et al., 2008*; *Olivera Harris et al., 2015*). We exploited its extremely high capacity for DNA synthesis to study MMR. A plasmid carrying an A:C mismatch, which is a

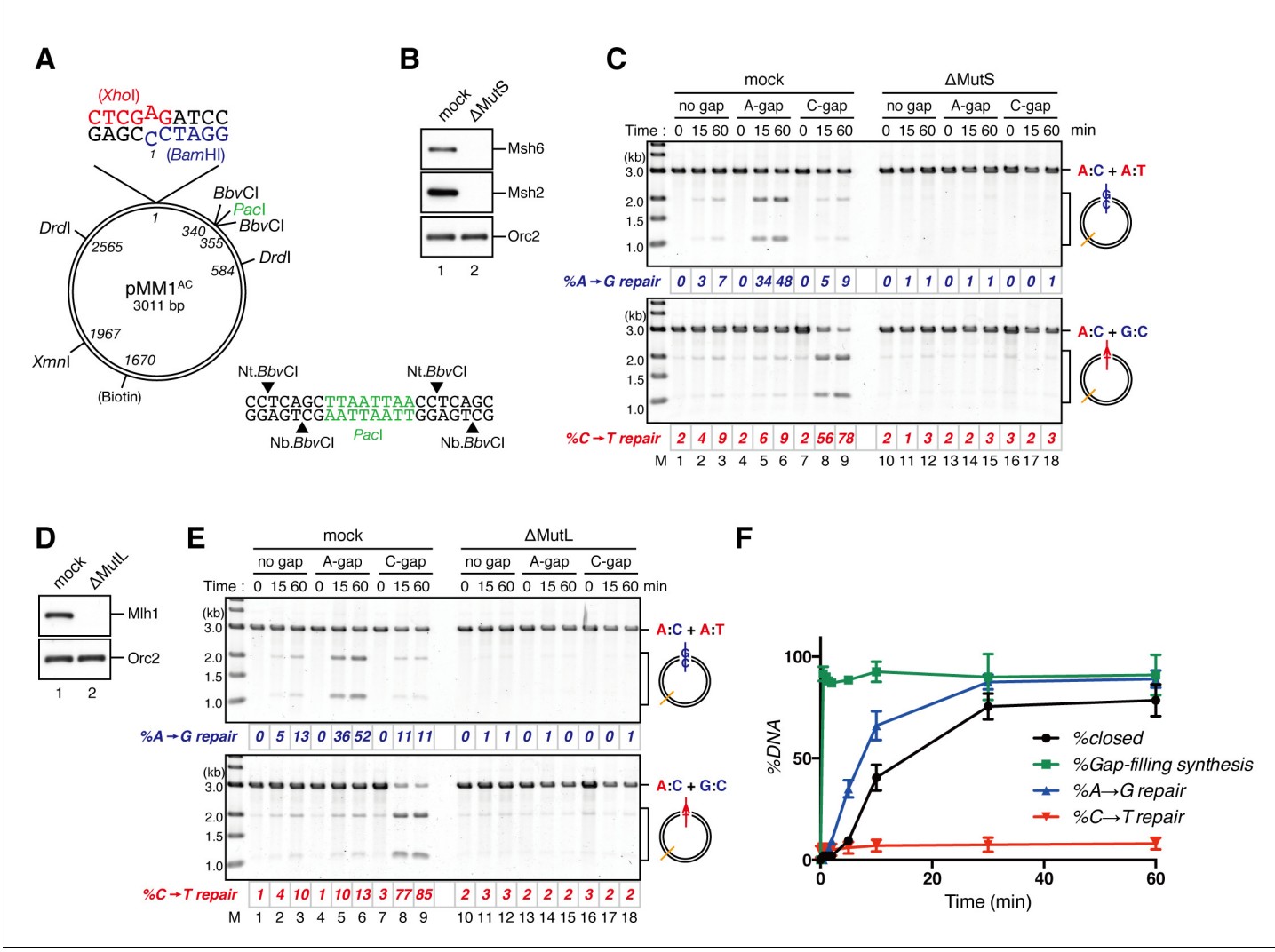

**Figure 1.** NPE promotes gap-directed MMR with efficient DNA synthesis at the gap site. (**A**) DNA substrates used in this study. The 3011-base pair plasmid DNA carries an A:C mispair (pMM1^AC), or an A:T base-pair (pMM1^AT) at position 1. Sequences surrounding an A:C mispair are designed so that the A:T or G:C repair product forms an *Xho*I or *Bam*HI cleavage site, respectively. Two *Bbv*CI nicking restriction enzyme sites were used to introduce a 15-nt single-strand gap. A *Pac*I site was placed within the gap. If needed, a biotin-dT modification was introduced at position 1670. (**B**) NPE was depleted using pre-immune antibodies (lane 1) or a mixture of Msh2 and Msh6 antibodies (lane 2). Immunoblots of the NPE samples (0.05 μl each) are shown. Orc2 served as a loading control. (**C**) Covalently closed (lanes 1–3 and 10–12), A-strand-gap (3' to the mismatch)-carrying (lanes 4–6 and 13–15) or C-strand-gap (5' to the mismatch)-carrying pMM1^AC (lanes 7–9 and 16–18) were incubated in NPE described in (**B**), and sampled at the indicated times. DNA was purified and digested with *Xmn*I and either *Bam*HI (upper, A to G repair) or *Xho*I (lower, C to T repair). *%repair* was calculated based on the percentage of *Xho*I or *Bam*HI sensitive DNA molecules. (**D**) NPE was depleted using pre-immune antibodies (lane 1) or Mlh1 antibodies (lane 2). Immunoblots of the NPE samples (0.05 μl each) are shown. (**E**) The MMR reaction in NPE described in (**D**). DNA was digested with *Xmn*I and either *Bam*HI (upper, A to G repair) or *Xho*I (lower, C to T repair). (**F**) Percentages of DNA synthesis at the gap, estimated by *Pac*I sensitivity (*%Gap-filling synthesis*), A to G repair (*%A→G repair*), C to T repair (*%C→T repair*) and closed circular molecules (*%closed*; *Figure 1—figure supplement 5*), calculated from two independent experiments including the one described in *Figure 1—figure supplement 6*, were plotted onto a graph. To calculate the *%Gap-filling synthesis*, DNA was digested with *Xmn*I and *Pac*I. The mean values were connected by lines. Error bars: ± 1 standard deviation (SD).

The following figure supplements are available for figure 1:

**Figure supplement 1.** Comparison of the MMR efficiencies between in crude *Xenopus* egg extracts and in NPE.

**Figure supplement 2.** Detection of tracts of DNA repair synthesis during gap-directed MMR in NPE.

**Figure supplement 3.** Characterization of xMsh2, xMsh6 and xMlh1 sera.

*Figure 1 continued on next page*

*Figure 1 continued*

**Figure supplement 4.** Comparison of the MMR efficiencies between nick-carrying and gap-carrying substrates.
**Figure supplement 5.** Kinetics of gap filling in NPE in the absence of a mismatch.
**Figure supplement 6.** Kinetics of gap filling and MMR on a mismatch-carrying DNA in NPE.

preferred MMR substrate in crude *Xenopus* egg extracts (*Varlet et al., 1990*), and a 15-nucleotide (nt) single-strand gap was constructed (*Figure 1A*). This plasmid was synthesized in vitro by second strand DNA synthesis after annealing of a primer carrying a 'C'-mismatch on single-stranded phage-mid DNA corresponding to the 'A'-strand. Since NPE contains a high concentration of Geminin, a specific inhibitor for assembly of pre-replicative complexes, no DNA replication initiates when DNA is incubated directly in NPE, while DNA synthesis from existing 3'-termini is active (*Walter et al., 1998*). NPE corrected the A:C mismatch by selectively editing the base on the gap-carrying strand (*Figure 1B–E*, see 'mock'), more efficiently than conventional crude *Xenopus* egg extracts (*Figure 1—figure supplement 1*). As seen previously (*Olivera Harris et al., 2015*), a gap could be placed on either 5' or 3' of the mismatch, and repair DNA synthesis occurred preferentially within a fragment containing the shorter path between the gap and the mismatch (*Figure 1—figure supplement 2*), indicating that NPE supports bidirectional MMR. Depletion of the Msh2-containing (MutSα/β) (*Figure 1B–C* and *Figure 1—figure supplement 3*) or Mlh1-containing complexes (MutLα/β/γ) (*Figure 1D–E* and *Figure 1—figure supplement 3*) abolished the repair reaction. Unlike other systems, a nick did not efficiently induce MMR (*Figure 1—figure supplement 4*). These results demonstrate that NPE efficiently recapitulates gap-directed, bi-directional MMR that is dependent on both MutSα/β and MutLα/β/γ. Some background repair observed in the absence of a gap could be due to either spontaneous base damages, which elicit base excision repair that in turn direct MMR (*Repmann et al., 2015*), or strand breaks that have occurred during handling of the substrate.

On a substrate that has no mismatch, a single-strand gap was sealed within 2 min in NPE, resulting in quick accumulation of closed circular plasmids (*Figure 1—figure supplement 5*). To understand the temporal relationship between gap filling and gap-directed MMR, we compared the kinetics of DNA synthesis at the gap site and repair of the mismatch simultaneously on the same substrate. As shown in *Figure 1F*, DNA synthesis at the gap site (estimated by digestion with *PacI* placed within the gap) was mostly completed within 30 s. However, this DNA synthesis did not covalently close the mismatch-carrying plasmid, suggesting that new strand breaks are introduced on the DNA (*Figure 1F* and *Figure 1—figure supplement 6*). The A to G MMR gradually progressed until ~30 min, and shortly after the mismatch correction, closed circular molecules were accumulated. These observations suggest that, in this gap-directed MMR model, efficient DNA synthesis at the gap site precedes mismatch correction involving degradation of the error-carrying strand. As suggested previously (*Varlet et al., 1996*), the 3'-terminus is likely used as a signal for MMR, rather than the initiation point of extensive strand removal.

## A gap-derived strand memory directs MMR even after gap filling

If the 3'-terminus generates a signal for strand discrimination, does the signal remain after the sealing of the gap, or is the 3'-terminus essential to initiate MMR? To answer this question, we set up a stepwise incubation assay that separates gap filling and mismatch recognition. We first incubated the MMR substrates in a MutSα/β-depleted (MutS-depleted) NPE, in which the gap should be filled and ligated without mismatch recognition, and then supplied MutSα/β by addition of fresh NPE (*Figure 2A*). In the MutS-depleted NPE, more than 80% of gap-carrying plasmids were converted into the closed-circular form within 1.5 min (*Figure 2B*, lane 9). However, ~40% of A to G MMR was still observed upon addition of fresh NPE at 2 min, at a time when at most, only ~14% strand discontinuities remained (*Figure 2B*, lane 10 and *Figure 2C*, lane 14). From 0.5 to 2 min, the MMR efficiency consistently exceeded the amounts of remaining strand discontinuities. These data suggest that the strand information derived from a gap is 'memorized' on DNA to induce subsequent MMR.

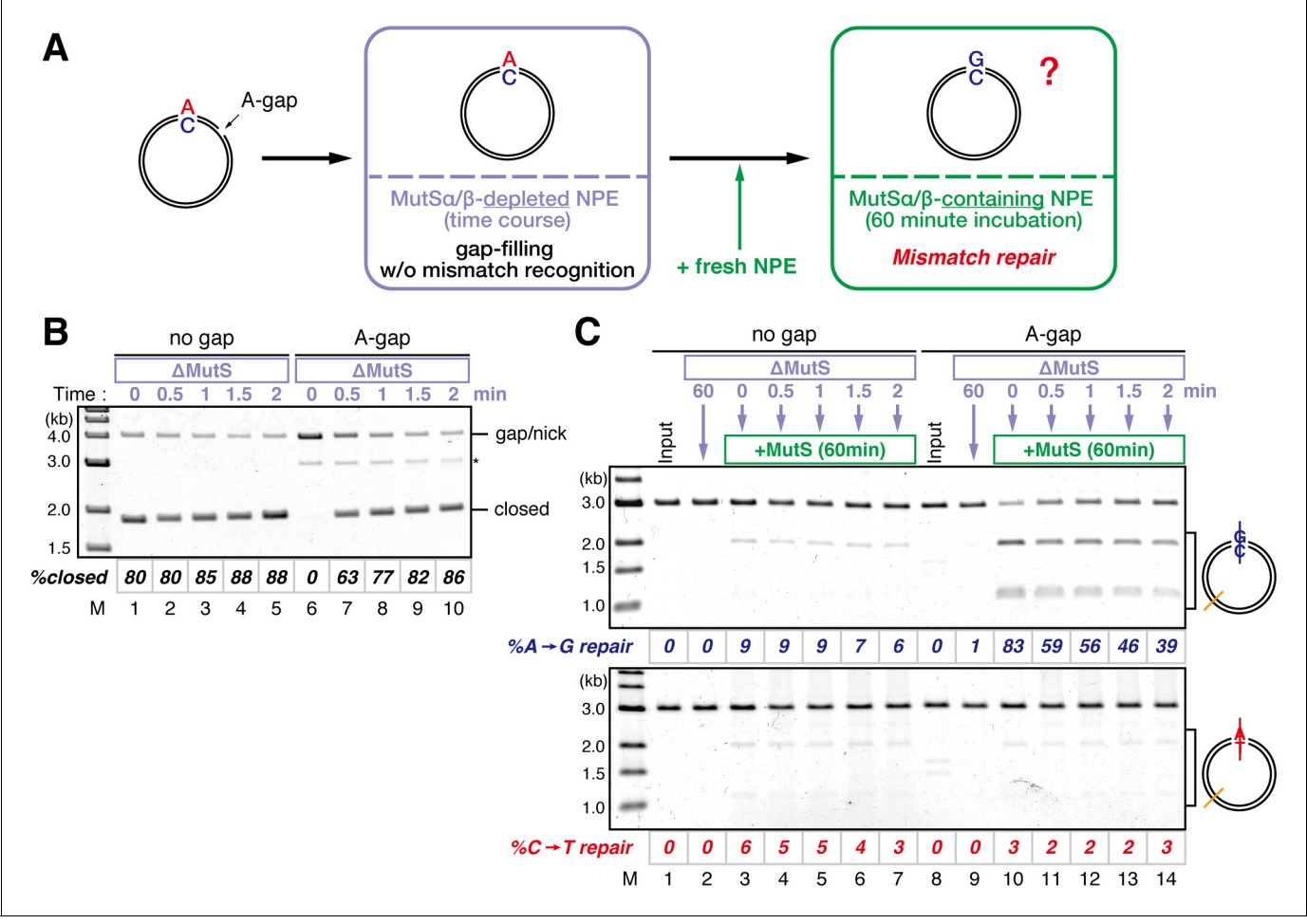

**Figure 2.** Strand memory derived from a gap directs MMR after the completion of gap filling. (**A**) Schematic diagram of the stepwise incubation assay. pMM1[AC] carrying a 15-nt gap was incubated in a MutS-depleted NPE to fill the gap without recognizing the mismatch. Subsequently, fresh NPE was added to an aliquot of the reaction to initiate MMR by supplying MutSα/β. After 60 min incubation, DNA was purified and the direction and the efficiency of repair were examined. (**B**) Kinetics of the gap-filling reaction in MutS-depleted NPE. Covalently closed (lanes 1–5) or A-strand-gap-carrying pMM1[AC] (lanes 6–10) was incubated in MutS-depleted NPE and sampled at the indicated times. (*) indicates linear DNA produced by contaminating endonuclease activity in Nt.*Bbv*CI. These linear molecules were excluded from the calculation of *%closed*. (**C**) Strand-specific MMR reaction after supplying MutSα/β. Aliquots were sampled at the indicated times, mixed with fresh NPE, and incubated for an additional 60 min. No repair was observed when the second NPE was omitted (lanes 2 and 9).

## Directional loading of PCNA bypasses requirement of a gap for strand-specific MMR in NPE

The gap-filling reaction in *Xenopus* egg extracts was PCNA-dependent (*Figure 3—figure supplement 1*). Therefore, at least one PCNA trimer must be involved in every gap filling, and it could retain the strand information after gap filling. If PCNA stores strand information for MMR, PCNA loaded on DNA in a specific orientation should direct MMR in NPE even in the absence of strand discontinuities. To test this idea, we established an in vitro PCNA-loading reaction using purified human RFC (hRFC) and PCNA (hPCNA), which can substitute for *Xenopus* PCNA (xPCNA) in the MMR reaction (*Figure 3—figure supplement 2*). A mismatch plasmid carrying an A-strand nick was bound to Sepharose beads through a site-specific biotin modification (see *Figure 1A*). hPCNA was loaded onto the immobilized plasmids in a nick-dependent, and therefore orientation-specific manner. The nick left on the DNA was sealed with the T4 DNA ligase. The hPCNA-DNA complex was then washed with 1 M KCl to remove hRFC, T4 ligase, and loosely associated hPCNA (*Figure 3A*).

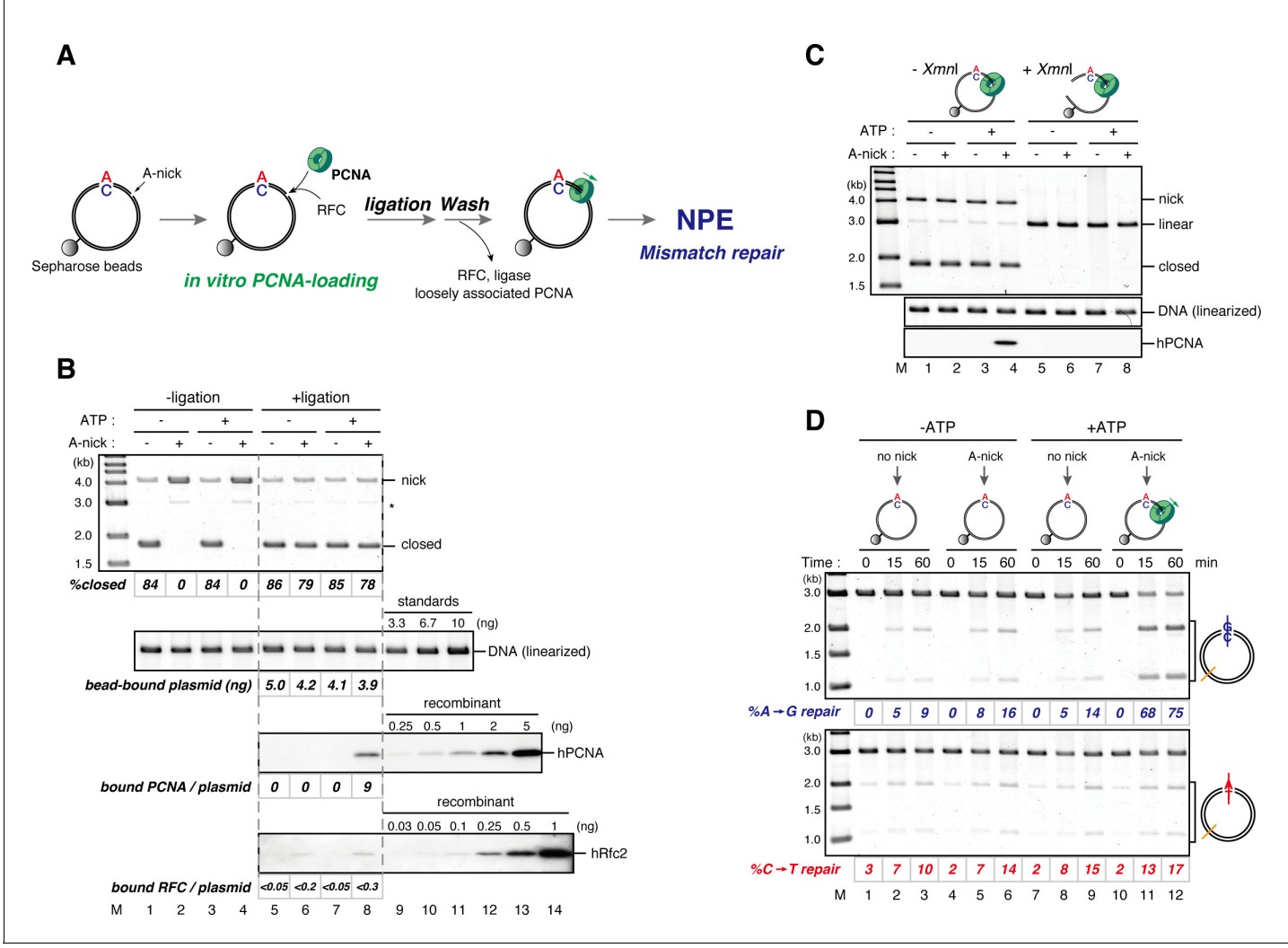

**Figure 3.** DNA-bound PCNA bypasses the requirement of a gap for strand-specific MMR in NPE. (**A**) Schematic diagram of the assay. Singly biotinylated pMM2^AC (a pMM1 derivative carrying only one *Bbv*CI site) carrying a nick was bound to Sepharose beads and incubated with recombinant hPCNA and hRFC. The nick was then ligated, and the complex was washed with a buffer containing 1 M KCl. The hPCNA-DNA complex was incubated in NPE to test whether MMR occurs. (**B**) In vitro PCNA-loading reaction. Untreated DNA (top), linearized DNA (middle, by *Xmn*I), and quantitative immunoblottings for hPCNA and hRfc2 (bottom) of samples from the in vitro PCNA-loading assay using covalently closed, and A-strand-nick-carrying pMM2^AC are presented. (*) indicates linear DNA, which was excluded from the calculation of *%closed*. (**C**) The hPCNA-DNA complexes described in (**B**) were split into two portions to test whether PCNA encircles DNA (**C**) and whether MMR occurs upon incubation in NPE (**D**). The one portion was treated with either control buffer, or buffer containing *Xmn*I whose cleavage site is located 1382 bp away from the PCNA entry point. DNA from the reaction (top), linearized DNA (middle, by *Xmn*I), and a hPCNA immunoblot (bottom) are shown. Since nick-carrying molecules were accumulated during incubation, the level of closed-circular molecules was lower than the original substrates shown in (**B**). (**D**) The other potion of the hPCNA-DNA complexes described in (**B**) was incubated in NPE. The MMR efficiencies were calculated as described in *Figure 1C*.

The following figure supplements are available for figure 3:

**Figure supplement 1.** Requirement of PCNA for gap-filling and nick-ligation reactions in *Xenopus* egg extracts.

**Figure supplement 2.** Characterization of recombinant hRFC and hPCNA.

Using fluorescent-based quantitative western blotting, we estimated that 4–10 hPCNA trimers reproducibly loaded on the DNA in a nick- and ATP-dependent manner (*Figure 3B*; see also *Figure 4*). Ligation of the nick was ~80% efficient. The DNA-bound hPCNA topologically encircled DNA with free-sliding ability, because linearization of DNA resulted in dissociation of essentially all hPCNA

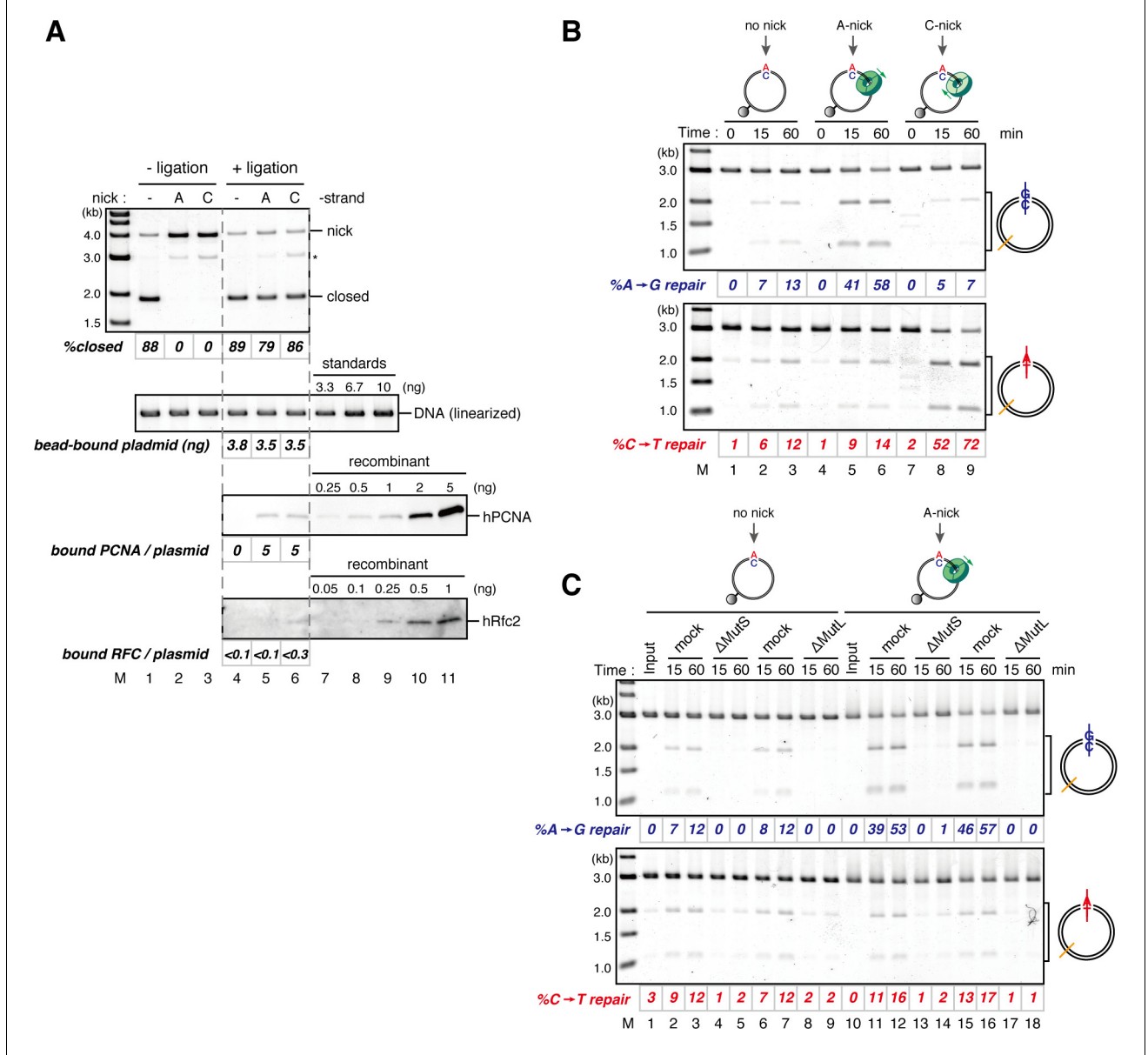

**Figure 4.** The orientation of PCNA loading determines strand specificity of MMR in NPE. (**A**) In vitro PCNA-loading reaction with pMM2$^{AC}$ carrying no nick, an A-strand nick or a C-strand nick. Untreated DNA (top), linearized DNA (middle, by *XmnI*), and quantitative immunoblots (bottom) are presented. (**B**) The hPCNA-DNA complexes described in (**A**) were incubated in NPE. (**C**) The hPCNA-DNA complexes (*Figure 4—figure supplement 2*) were incubated in mock-treated, MutS-depleted or MutL-depleted NPE described in *Figure 4—figure supplement 2*.

The following figure supplements are available for figure 4:

**Figure supplement 1.** Requirement of hPCNA and hRFC in PCNA-directed MMR in NPE.

**Figure supplement 2.** Requirement of the MutS and MutL complexes in PCNA-directed MMR in NPE.

(*Figure 3C*). As PCNA was loaded from the A-strand nick in this case, A to G MMR was predicted. When the hPCNA-DNA complex was transferred into NPE, ~75% of the A bases were specifically repaired into G bases (*Figure 3D*). Such a high repair efficiency could not be explained by the remaining nicks, which could contribute to at most 20% of MMR.

If PCNA directs strand-specific MMR, PCNA loaded onto DNA in the opposite orientation should invert the strand to be repaired. Critically, the hPCNA-DNA complex prepared with A-nick DNA directed ~60% of A to G MMR, and the complex prepared with C-nick DNA directed ~70% of C to T MMR in NPE (*Figure 4A and B*). When either PCNA or RFC was omitted from the reaction, most of the strand-specific repair was inhibited (*Figure 4—figure supplement 1*). The PCNA-directed MMR reaction was dependent on both MutSα/β and MutLα/β/γ (*Figure 4C* and *Figure 4—figure supplement 2*). These results indicate that directional loading of hPCNA is sufficient to induce strand-specific MMR in the absence of strand discontinuities in NPE.

## The MutS complexes inhibit Rfc3-dependent PCNA unloading

The above results suggest that DNA synthesis at the gap site leaves PCNA on DNA as the strand memory, which can be subsequently used for MMR. Consistent with this idea, a significant portion of nick ligation was independent of PCNA, explaining why a nick did not induce MMR as efficiently as a gap in NPE (*Figure 3—figure supplement 1*). In the absence of MutSα/β, the strand information was gradually lost after the disappearance of strand discontinuities (*Figure 2*), suggesting that PCNA is unloaded from DNA within this time window. To examine PCNA dissociation in NPE, we loaded hPCNA on DNA, incubated it in NPE, and quantified the remaining hPCNA using an hPCNA specific monoclonal antibody that does not cross-react with xPCNA (*Figure 5—figure supplement 1*). This method distinguishes retention of hPCNA from sticking of xPCNA. In the mock-treated NPE, hPCNA was quickly dissociated from a homoduplex DNA within a few minutes, while dissociation was attenuated in the NPE depleted of xRfc3, indicating that PCNA is removed from DNA mostly through an active unloading process requiring the Rfc3-containing complex(es) (*Figure 5*). Due to the unavailability of appropriate antibodies, the specific Rfc3-containing complex responsible for PCNA unloading could not be identified.

Amount of DNA-bound PCNA should correlate with the MMR capability. In addition, since DNA-bound MutSα/β can interact with PCNA (*Flores-Rozas et al., 2000*; *Bowers et al., 2001*; *Lau and Kolodner, 2003*; *Iyer et al., 2008*), MutSα/β might affect dynamics of PCNA retention. We therefore examined hPCNA retention and the MMR capability in the presence or absence of MutSα/β (*Figure 6A*). In the MutS-depleted NPE, hPCNA was again quickly unloaded from mismatch-carrying DNA (*Figure 6B* and *Figure 6—figure supplement 1*). Concurrent with PCNA unloading, the PCNA-directed MMR capability rapidly dropped, leaving only ~10% of repairable molecules at 5 min (*Figure 6C*, lane 5). In contrast, in the MutL-depleted NPE (containing endogenous MutSα/β but deficient for MMR), even at 30 min, more than one hPCNA trimer on an average still remained on DNA, and ~20% of mismatches were still repairable (*Figure 6C*, lane 14). The difference in PCNA retention and the kinetics of repair was not due to remaining nicks on the DNA, because neither MutS- nor MutL-depletion significantly affected the level of nick-carrying molecules (*Figure 6—figure supplement 2*, see also *Figure 7*). These results suggest that mismatch-bound MutS complexes interfere with unloading of PCNA to retain strand information for MMR.

## The MutS complexes retain strand memory derived from a gap

To test whether MutSα/β also maintain the MMR capability derived from a gap, we repeated the stepwise incubation experiments presented in *Figure 2*, but used MutL-depleted NPE as the first NPE this time. At least 80% of the gaps were filled in all conditions at 2 min (*Figure 7A and B*, and *Figure 7—figure supplement 1*). In either MutS-depleted or MutS/MutL-depleted NPE, more than 40% of mismatches were still repairable upon addition of fresh NPE at 2 min, confirming the ~2 min strand memory seen in *Figure 2* (*Figure 7C and D*). Strikingly, in the MutL-depleted NPE, >60% of mismatches were repairable at 10 min, and ~35% were still repairable even at 30 min. Using one-phase exponential decay fitting, which could explain most data with reasonable quality ($R^2 > 0.8$), we estimated the half-life of the strand memory in the absence of MutSα/β to be approximately 2 min, and to be approximately 40 min in the presence of MutSα/β (*Table 1*). These results strongly suggest that the MutSα/β-dependent mechanism maintains the strand information by inhibiting unloading of PCNA that is loaded during DNA synthesis at the gap site.

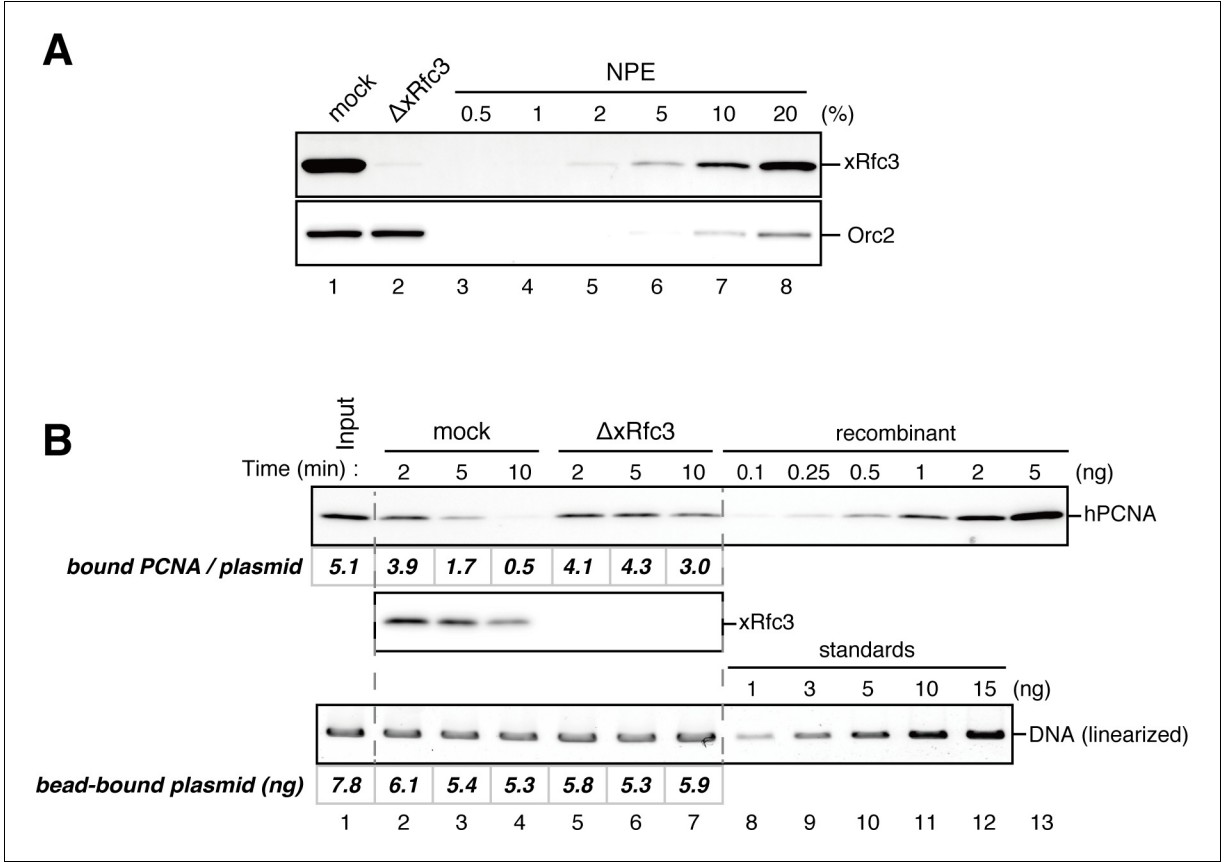

**Figure 5.** Rfc3 is required for the unloading of PCNA from closed-circular DNA. (**A**) Depletion efficiency of xRfc3 from NPE. The depletion efficiency was estimated to be >98%. (**B**) hPCNA loaded onto immobilized pMM2$^{AT}$ (carrying no mismatch) was incubated in NPE described in (**A**). The efficiency of nick ligation in the PCNA loading reaction was estimated to be ~89%.

The following figure supplement is available for figure 5:

**Figure supplement 1.** Specific detection of hPCNA by a monoclonal antibody.

## The PIP motif in MutSα is required for efficient retention of strand information and PCNA

To address whether the PIP motif located at the N-terminus of Msh6 is required for retention of DNA-loaded PCNA and strand information, we purified recombinant xMutSα harboring alanine substitutions in the PIP motif (xMutSα$^{PIP}$; *Figure 8A and B*). In good agreement with previous reports (*Kleczkowska et al., 2001*; *Iyer et al., 2008*; *Bowen et al., 2013*), when no stepwise incubation was involved, the gap-directed MMR on the 3.0 kb substrate was largely restored by xMutSα$^{PIP}$ in MutS-depleted NPE (*Figure 8—figure supplement 1*), indicating that in this experimental setup the PIP function is not important for MMR. Mismatch binding of xMutSα$^{PIP}$ was not significantly compromised (*Figure 8—figure supplement 1*). However, xMutSα$^{PIP}$ was almost inert for PCNA retention. We repeated the PCNA-unloading assay using MutS/MutL-depleted NPE supplemented with xMutSα$^{WT}$ or xMutSα$^{PIP}$ (*Figure 8C*). Although xMutSα$^{WT}$ attenuated PCNA unloading, xMutSα$^{PIP}$ did not exhibit detectable retention of PCNA on mismatched DNA, compared to the buffer control (*Figure 8D*). In the gap-derived strand memory assay, xMutSα$^{WT}$ restored retention of strand memory in the MutS/MutL-depleted NPE without affecting the gap-filling reaction ($t_{1/2}$ = 25.5 min; *Figure 9*, *Figure 9—figure supplement 1*, and *Table 2*). In contrast, only modest restoration of the retention of strand information was seen with xMutSα$^{PIP}$ ($t_{1/2}$ = 12.5 min). These results indicate that the PIP motif is required for inhibition of PCNA unloading and largely responsible for strand memory

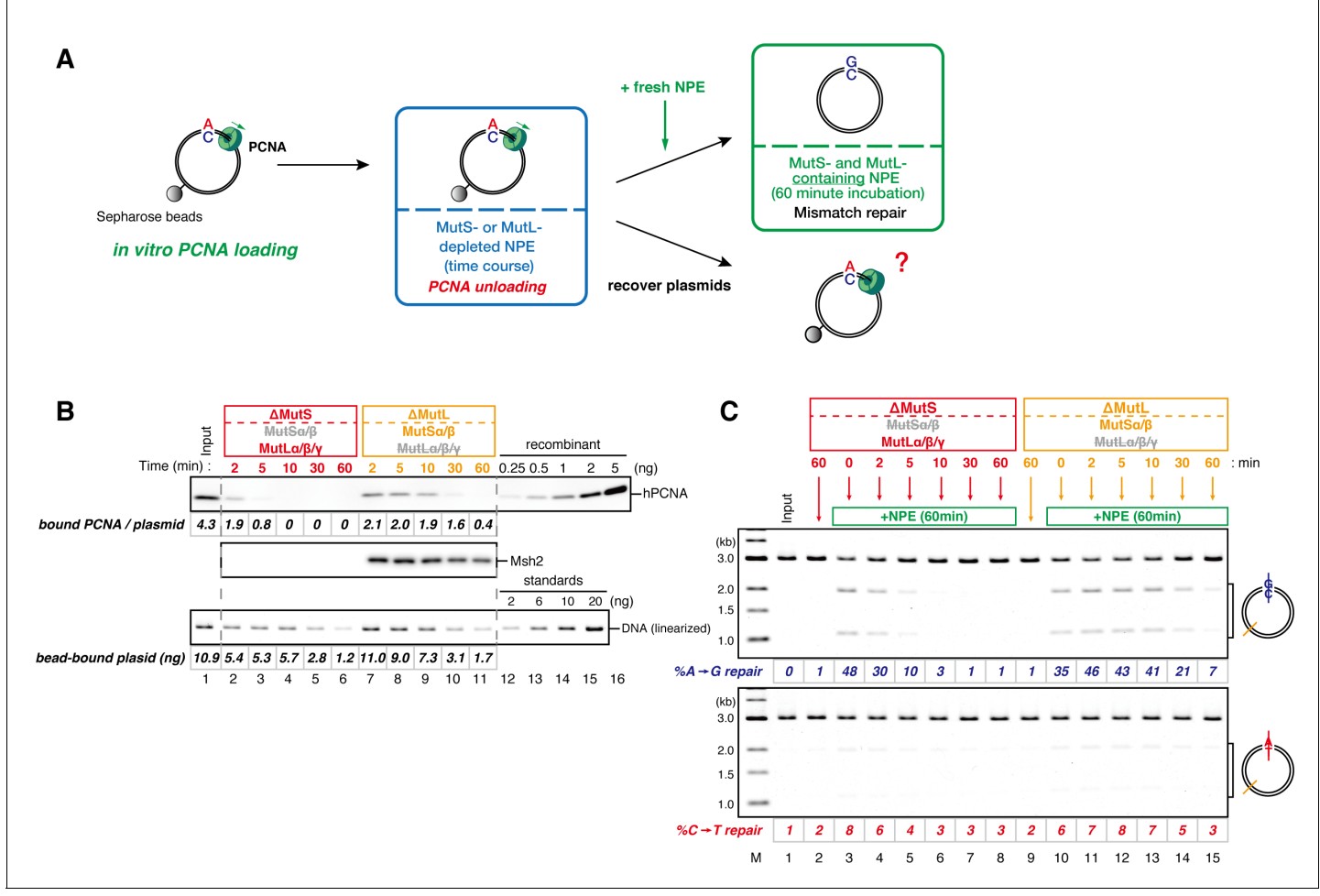

**Figure 6.** The MutS complexes inhibit Rfc3-dependent PCNA unloading in NPE. (**A**) Schematic diagram of the stepwise incubation assay. After the in vitro PCNA-loading reaction, hPCNA-DNA complexes were incubated in either MutS- or MutL-depleted NPE. The samples were split into two portions, fresh NPE was added to one portion to measure MMR, and DNA-bound proteins were recovered from the other portion to quantify hPCNA. (**B**) hPCNA loaded onto immobilized pMM2$^{AC}$ carrying an A-strand nick was incubated in NPE depleted of either MutSα/β or MutLα/β/γ. The efficiency of nick ligation in the PCNA loading reaction was estimated to be ~72%. The amounts of DNA-bound hPCNA are shown. See *Figure 6—figure supplement 1* for the depletion efficiencies (MutS: >98%, MutL: >98%). (**C**) Kinetics of the PCNA-directed MMR capability in the presence or absence of MutSα/β. Aliquots were sampled from the reaction described in (**B**) and mixed with fresh NPE.

The following figure supplements are available for figure 6:

**Figure supplement 1.** Depletion efficiencies of Msh2 and Mlh1 from NPE.

**Figure supplement 2.** Effect of MutS- and MutL-depletion on the level of closed-circular DNA molecules.

retention, yet a PIP-independent mechanism also contributes to the retention of strand information derived from a gap.

In yeast, the N-terminal region (NTR) of Msh6 not only carries a canonical PIP motif but also contains some cryptic PCNA binding sites (*Shell et al., 2007*). To examine whether the NTR of Msh6 is responsible for the weak retention of strand memory, we purified xMutSα lacking the entire NTR of Msh6 (Δ1–323; xMutSα$^{ΔN}$). As with xMutSα$^{PIP}$, xMutSα$^{ΔN}$ did not show significant defects in gap-directed MMR and in mismatch binding (*Figure 8—figure supplement 1*), and was essentially deficient in PCNA retention (*Figure 8C and D*). Interestingly, the xMutSα$^{ΔN}$ complex partially restored the strand memory reaction, with an estimated half-life of 12.3 min, although it did not affect the kinetics of the sealing of the gap in the first NPE (*Figure 9* and *Table 2*). We concluded that the

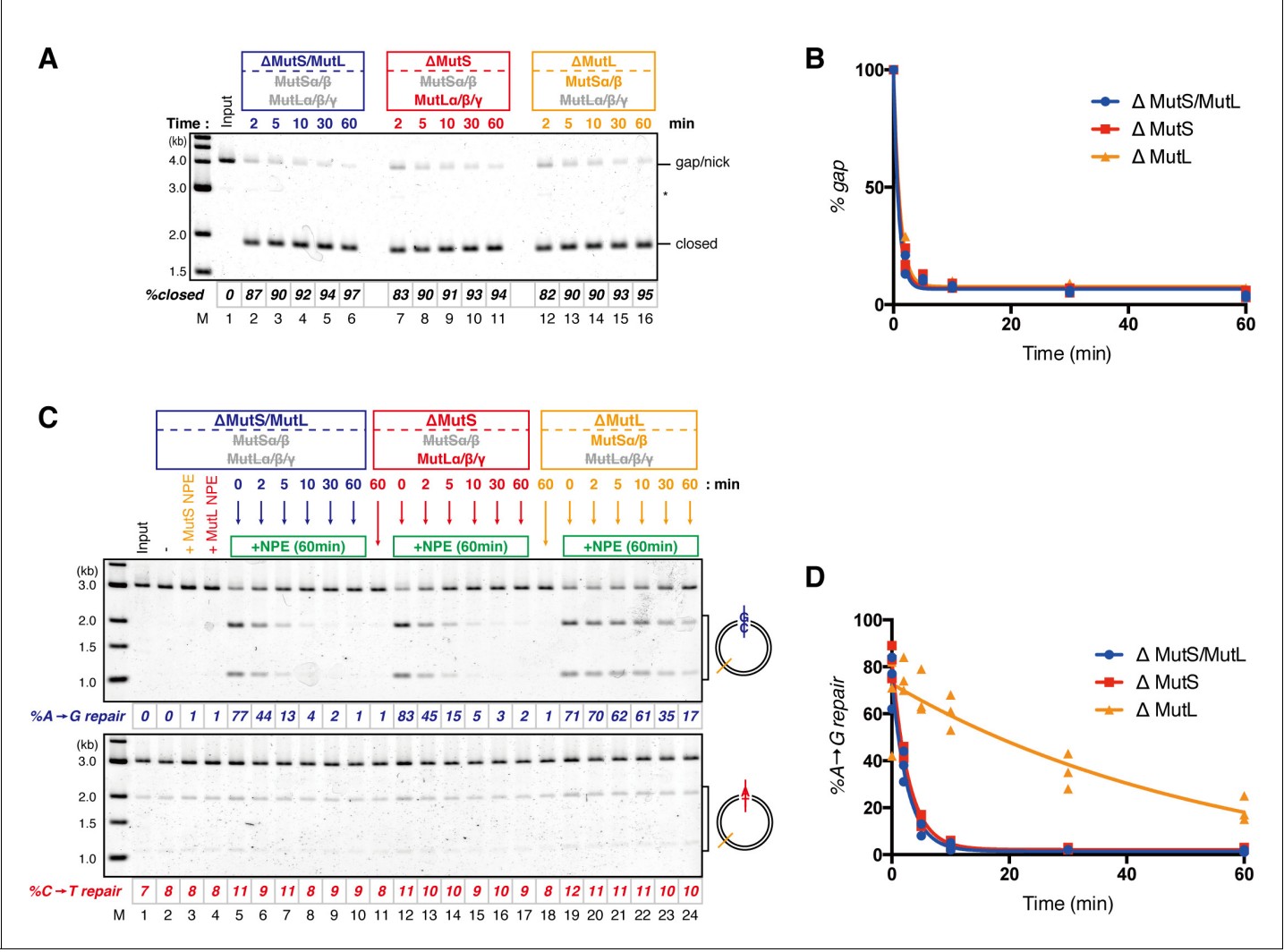

**Figure 7.** The MutS complexes retain strand memory derived from a gap in NPE. (A) The gap-filling reaction of A-strand-gap-carrying pMM1^AC in NPE described in *Figure 7—figure supplement 1*. (B) The efficiencies of the gap-filling reaction were calculated from three independent experiments including the one described in (A) and plotted on a graph. The lines represent one-phase decay fitting of the data. See *Table 1* for the fitting parameters. (C) Strand-specific MMR reaction after supplying fresh NPE. (D) The efficiencies of MMR were calculated and presented as described in (B).

The following figure supplement is available for figure 7:

**Figure supplement 1.** Depletion efficiencies of Msh2 and Mlh1 from NPE.

retention of strand memory is mediated largely by the PIP motif in MutSα, and there is a partial strand memory retention that is independent of the NTR of Msh6.

## Discussion

Identification of the newly synthesized DNA strand is vital to post-replicative mismatch repair. Since DNA replication generates identical copies of parental DNA, the strand discrimination reaction in MMR necessarily requires direct or indirect interplay with the replication complex (*Wagner and Meselson, 1976*). Because of the asymmetric nature of its binding to DNA, its critical roles in an early step of MMR and in the activation of MutLα, PCNA is presumed to be the mediator of the interplay between the replication complex and MMR (*Umar et al., 1996*; *Chen et al., 1999*; *Pavlov et al., 2003*; *Dzantiev et al., 2004*; *Kadyrov et al., 2006*; *Pluciennik et al., 2010*;

**Table 1.** Fitting parameters for the gap-derived strand memory experiment.

| | %gap [95% Confidence Interval] | | | %A→G repair [95% Confidence Interval] | | |
|---|---|---|---|---|---|---|
| | $t_{1/2}$ (min) | %gap at 0 min | $R^2$ | $t_{1/2}$ (min) | %repair at 0 min | $R^2$ |
| ΔMutS/MutL | 0.60 [0.51−0.73] | 100 [96.7−103.3] | 0.99 | 1.87 [1.52−2.45] | 74.8 [68.6−80.7] | 0.97 |
| ΔMutS | 0.68 [0.58−0.81] | 100 [96.4−103.5] | 0.99 | 2.00 [1.76−2.31] | 82.8 [79.0−86.6] | 0.99 |
| ΔMutL | 0.75 [0.66−0.88] | 100 [96.5−103.5] | 0.99 | 39.6 [12.3− +∞] | 72.8 [63.6−81.9] | 0.80 |

Parameters for the one-phase exponential decay fitting of the data described in **Figure 7B and D** are presented. %gap: percentage of remaining gaps (100 - %closed), $t_{1/2}$: half-life, $R^2$: coefficient of determination. n = 3. Curve fitting was carried out using the GraphPad Prism 6 software (GraphPad Software, CA, USA)

Hombauer et al., 2011b; Peña-Diaz and Jiricny, 2012; Georgescu et al., 2015; Kunkel and Erie, 2015). In this study, we provided experimental evidence that DNA-loaded PCNA indeed directs strand-specific MMR in the absence of strand discontinuities. We further discovered that MutSα inhibits PCNA unloading using its PIP motif on the N-terminus of Msh6, and that this reaction significantly extends the temporal window during which MMR is possible. Our results thus uncovered a

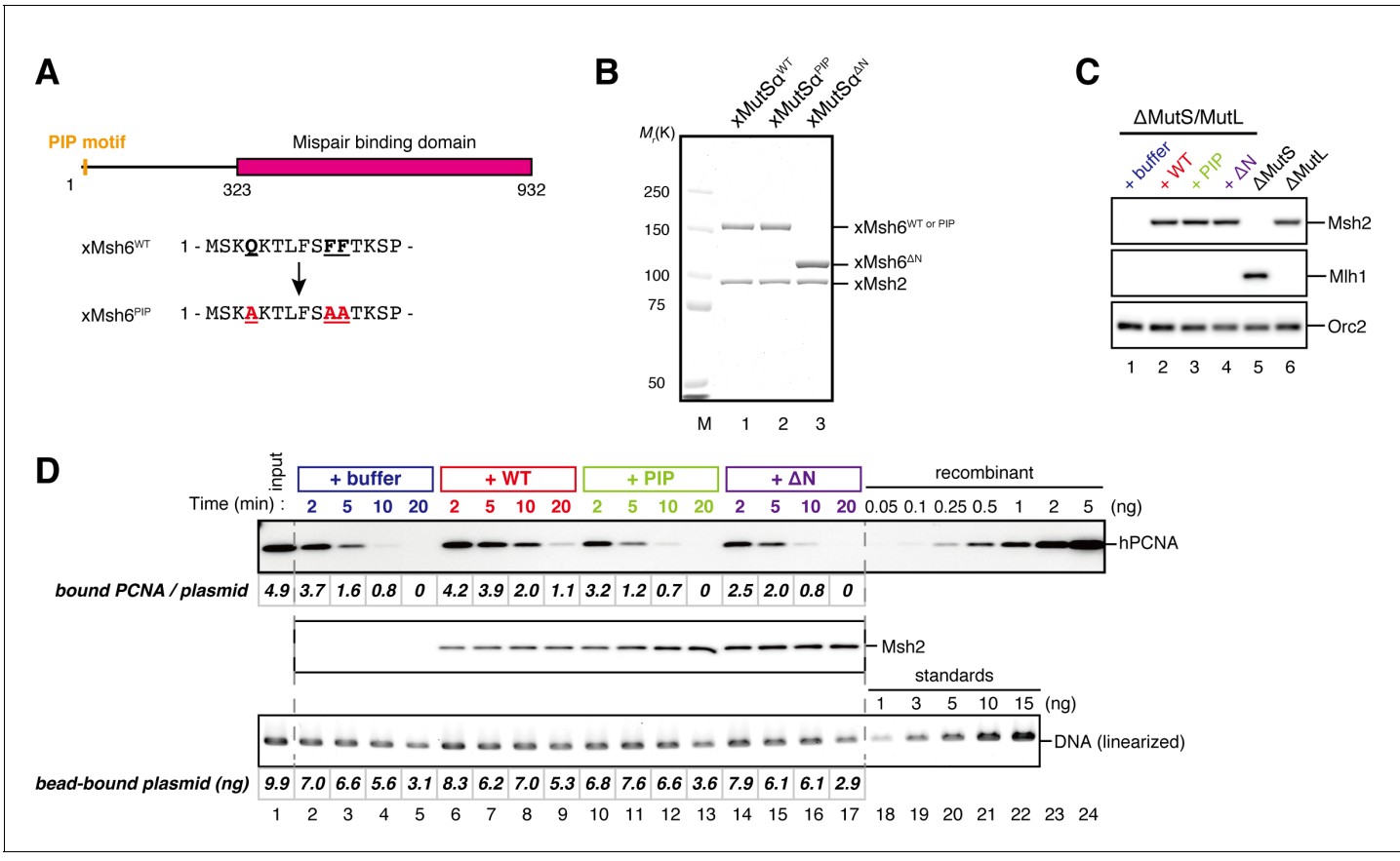

**Figure 8.** The PIP-motif of xMutSα is important for retention of PCNA. (A) The primary structure of Msh6 and the mutation sites in the PIP motif are presented. (B) 1 μg each of purified xMutSα complexes were separated by SDS-PAGE and stained with Coomassie brilliant blue R-250. (C) NPE was depleted of both MutS and MutL (lanes 1–4), MutS (lane 5), or MutL (lane 6) and supplemented with either control buffer, 630 nM of xMutSα[WT], xMutSα[PIP], or xMutSα[ΔN]. Immunoblots of each NPE are shown. (D) hPCNA loaded onto immobilized pMM2[AC] carrying an A-strand nick was incubated in NPE described in (C). The efficiency of nick ligation in the PCNA loading reaction was estimated to be ~75%.

The following figure supplement is available for figure 8:

**Figure supplement 1.** Requirement of the PIP-containing N-terminal domain of MutSα in MMR and mismatch binding.

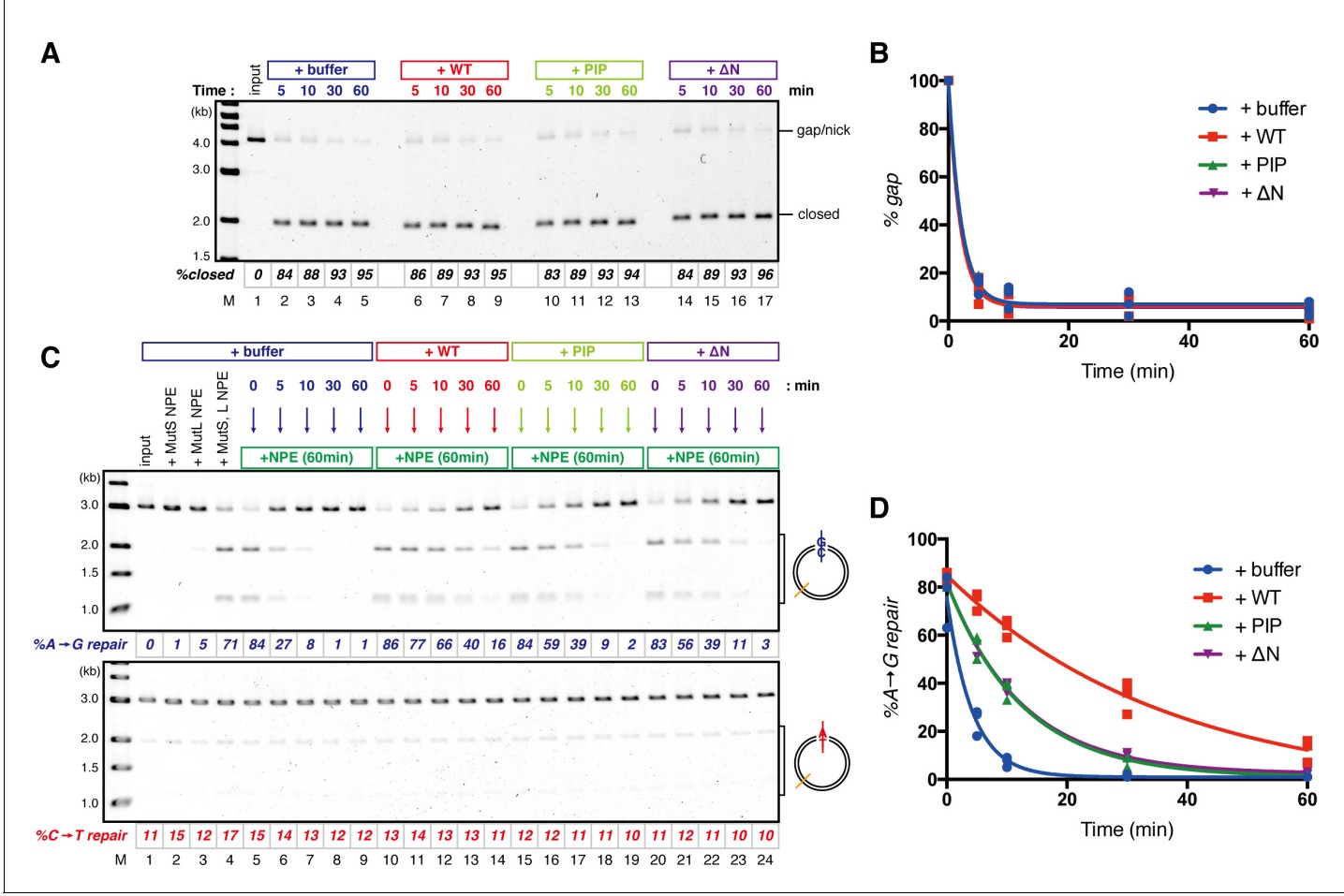

**Figure 9.** The PIP-motif of xMutSα is important for retention of the MMR capability derived from a gap. (A) Kinetics of the gap-filling reaction in MutS/MutL-depleted NPE containing 630 nM of xMutSα$^{WT}$, xMutSα$^{PIP}$, or xMutSα$^{ΔN}$. (B) The efficiencies of the gap-filling reaction were calculated from three independent experiments including the one shown in (A) and plotted onto a graph. Lines: one-phase decay fitting. See *Table 2* for the fitting parameters. (C) Kinetics of MMR after supplying fresh NPE to the reaction described in (A). Addition of neither MutS-depleted nor MutL-depleted NPE alone, but both restored MMR in MutS/MutL-depleted NPE (lanes 2–4), showing both MutSα/β and MutLα/β/γ were functionally depleted. (D) The efficiencies of MMR were calculated and presented as described in (B).

The following figure supplement is available for figure 9:

**Figure supplement 1.** Depletion efficiencies of Msh2 and Mlh1 from NPE.

novel mechanism in which interplay between PCNA and MutSα maintains the post-replicative temporal window for MMR.

NPE has been extensively used as a physiological model system that recapitulates various DNA repair reactions coupled with DNA synthesis (*Räschle et al., 2008*; *Olivera Harris et al., 2015*). As seen in other systems, we observed efficient bidirectional MMR in NPE (frequently >80% efficient). Curiously, we noticed some difference between NPE and other systems. Both in reconstitution systems and in human cell extracts, 5'-nick directed MMR can occur in the absence of MutLα, whereas 3'-nick directed MMR depends on MutLα (*Drummond et al., 1996*; *Genschel and Modrich, 2003*; *Dzantiev et al., 2004*; *Constantin et al., 2005*; *Bowen et al., 2013*). In contrast, not only 3'-gap directed but also 5'-gap directed MMR was dependent on MutL complexes in NPE (*Figure 1B–E*). Although this could possibly reflect difference in the 5' to 3' strand excision mechanism between species, we assume that the difference largely come from highly efficient DNA synthesis and ligation in NPE; quick sealing of a gap would necessitate MutLα-dependent incision for MMR even when a gap was placed on the 5'-side of the mismatch. Interestingly, DNA synthesis at the 3'-gap clearly

**Table 2.** Fitting parameters for the gap-derived strand memory experiment with MutSα mutants

| | %gap [95% Confidence Interval] | | | %A→G repair [95% Confidence Interval] | | |
|---|---|---|---|---|---|---|
| | $t_{1/2}$ (min) | %gap at 0 min | $R^2$ | $t_{1/2}$ (min) | %repair at 0 min | $R^2$ |
| +buffer | 1.44 [0.51−0.73] | 100 [95.2−104.7] | 0.99 | 2.94 [2.32−3.98] | 75.7 [69.2−82.2] | 0.97 |
| +xMutSα$^{WT}$ | 1.35 [0.58−0.81] | 100 [95.1−104.9] | 0.99 | 25.5 [17.6−46.1] | 84.8 [80.5−89.2] | 0.98 |
| +xMutSα$^{PIP}$ | 1.48 [1.18−2.00] | 100 [95.4−104.6] | 0.99 | 12.5 [10.9−14.7] | 81.9 [78.5−85.4] | 0.99 |
| +xMutSα$^{ΔN}$ | 1.43 [1.14−1.93] | 100 [95.6−104.4] | 0.99 | 12.3 [11.3−13.6] | 81.9 [79.8−84.0] | 1.00 |

Parameters for the one-phase exponential decay fitting of the data described in **Figure 9B and D** are presented. *%gap*: percentage of remaining gaps (100 - *%closed*), $t_{1/2}$: half-life, $R^2$: coefficient of determination. *n* = 3.

precedes MMR (**Figure 1F**), indicating that strand degradation does not progressively occur from the 3'-terminus of the gap. PCNA molecules loaded at the 3' terminus, perhaps more than one trimer, could be used by both MMR and DNA polymerases without significantly interfering with each other (see below). Consistently, a nick is suggested to function as a strand signal rather than an entry point for exonucleases in crude extracts of *Xenopus* eggs (**Varlet et al., 1996**).

Our stepwise incubation experiments demonstrated that MMR is possible even after the complete sealing of the gap (**Figure 2**). Three lines of evidence indicate that the molecule responsible for memorizing strand information is PCNA. First, directional loading of PCNA efficiently bypassed the need for strand discontinuities in strand-specific MMR (**Figure 3** and **4**). Second, the amount of DNA-bound PCNA correlated with the MMR capability (**Figure 6**). Third, MutSα maintained the MMR capability derived from a gap largely through the PIP motif (**Figure 9**). The fact that PCNA was essential for the gap-filling reaction in NPE also supports this conclusion, because at least one PCNA trimer must be involved in DNA synthesis at the gap. It is unlikely that RFC determines the strand specificity of MMR in our experiments, because amounts of RFC remained on DNA were insufficient to explain the efficiency of MMR. Based on the above evidence, we suggest that PCNA would function as the strand discrimination signal for MMR after the disappearance of local strand discontinuities during DNA replication. A series of seminal studies from the Modrich lab has led to the model in which DNA-bound PCNA activates the MutLα nicking endonuclease to initiate strand specific MMR (**Kadyrov et al., 2006**; **2007**; **Pluciennik et al., 2010**; **2013**). Our data are consistent with this prevailing model, since 1) DNA-bound PCNA induced strand-specific MMR in NPE, and 2) PCNA-directed MMR in NPE was completely dependent on the MutL complexes. As gap-directed MMR in NPE was bidirectional and preferentially degrades the DNA strand through the shortest path from the gap to the mismatch, we assume that the PCNA-directed MMR would function in the same manner. If so, the strand excision complex presumably built upon a specific face of PCNA would catalyze strand degradation in either direction depending on the relative orientation toward a mismatch. How this can be achieved is an important next question, and the system established here will be a useful tool to investigate this mechanism. However, since our plasmid substrates carry multiple PCNA molecules with free sliding ability, we currently do not exclude a possibility that PCNA-directed MMR catalyzes only a specific direction of strand excision, and PCNA molecules localized on a specific side toward a mismatch are preferentially used for MMR.

Unexpectedly, the 'strand memory' experiments revealed that MutSα/β significantly extend the duration of strand memory. Our data clearly indicate that this retention involves inhibition of PCNA unloading. A recent study showed that DNA polymerase δ captures DNA-bound PCNA which is otherwise quickly unloaded by the clamp-unloading activity of RFC (**Hedglin et al., 2013**). It would be possible that MutSα inhibits PCNA unloading in a similar manner; binding of the PIP motif onto PCNA may physically prevent the engagement of PCNA with RFC. Since in vivo PCNA unloading during S-phase largely depends on the Elg1-containing RFC-like complex (Elg1-RLC) both in humans and in yeast (**Kubota et al., 2013**; **Lee et al., 2013**; **Shiomi and Nishitani, 2013**; **Kubota et al., 2015**), the PIP-motif in MutSα may also limit the access of Elg1-RLC to DNA-bound PCNA. Detailed understanding of the molecular mechanism of this inhibition of PCNA unloading must await reconstitution of this reaction in vitro.

Interestingly, the PIP motif and the mispair-binding domain are connected by a long (~300 a.a.) linker that is predicted to be disordered in yeast (**Shell et al., 2007**). Although a significant portion

of this linker in human MutSα seems to adopt a globular conformation (*Iyer et al., 2008*), MutSα might be able to retain PCNA that is located far from MutSα using this long 'PIP-arm', possibly even over nucleosomes. In this scenario, strand removal would initiate when MutLα reaches to PCNA that is retained by the PIP-arm of MutSα. This could be accomplished either by sliding of MutSα-MutLα complexes along the DNA contour, or through loading of multiple MutLα molecules on DNA (*Hombauer et al., 2011a*; *Qiu et al., 2015*). Interaction between PCNA that is retained by MutSα and MutLα may invoke a specific strand degradation pathway, because a recent study in yeast has suggested that recruitment or retention of PCNA by MutSα around mismatches activates a specific strand removal pathway that is independent of Exo1 (*Goellner et al., 2014*). As seen for MutSα, we predict that MutSβ may inhibit PCNA unloading as well, since the NTR of yeast Msh6 and Msh3 are inter-exchangeable without significant deleterious effects (*Shell et al., 2007*). However, because NTRs of MutSα and MutSβ likely possess different characteristics such as the PWWP domain found only in vertebrate Msh6 (*Clark et al., 2007*; *Laguri et al., 2008*; *Iyer et al., 2010*; *Li et al., 2013*), this point will need rigorous investigation.

To our surprise, xMutSα$^{PIP}$ and xMutSα$^{ΔN}$ retained partial strand memory. A possibility would be that, although our PCNA unloading experiment failed to detect attenuation of PCNA unloading with these mutants (*Figure 8D*), they might have the ability to maintain DNA-bound PCNA that is below our detection limit. Considering the fact that we quantified PCNA after several washing steps, our quantification almost certainly underestimated the number of PCNA molecules on DNA. Alternatively, PCNA and/or a gap could induce structural alteration of MutSα (and possibly other chromatin binding factors), to maintain the strand information. How MutSα retains strand memory independently of the NTR will be an interesting question for future studies.

From the data presented in this work, we propose a model wherein two different modes of strand memory contribute to MMR (*Figure 10*). After the completion of local DNA synthesis, *e.g.* after ligation of Okazaki fragments, PCNA remains on DNA for a certain period until unloaded by an Rfc3-containing complex(es) ($t_{1/2}$ = ~2 min in NPE; *Figure 10*, [2]). The MMR system utilizes such PCNA as strand memory. However, if PCNA is not available closely adjacent to mismatches, MutSα (and possibly MutSβ also) retains available PCNA with its long 'PIP-arm', to maintain strand memory for subsequent MMR (*Figure 10*, [3]). This mode of memory survives for >30 min in NPE, and perhaps even longer in cells. In addition to the PIP-dependent tethering of MutSα/β to the replication fork (*Kleczkowska et al., 2001*; *Hombauer et al., 2011a*; *Haye and Gammie, 2015*), we suggest that the PIP motif would contribute to the replication fidelity through this mode of strand memory. Yeast studies have shown that the PIP motif has an assistive but not essential role in MMR (*Clark et al., 2000*; *Flores-Rozas et al., 2000*). The PIP-deficient MutSα complexes were proficient in repairing a mismatch in our in vitro MMR system (*Figure 8—figure supplement 1*), and similar observations were already reported both in human cell extracts and in a yeast in vitro reconstitution system (*Kleczkowska et al., 2001*; *Iyer et al., 2008*; *Bowen et al., 2013*). Therefore, a simple MMR reaction involving a mismatch and a closely-placed strand-discrimination signal, either PCNA or a strand discontinuity, would not require retention of PCNA by MutSα. However, when a stepwise incubation, which mimics transient arrest of the MMR reaction, was involved, the PIP motif became important for MMR (*Figure 9*). We speculate that inhibition of PCNA unloading is needed for MMR only under specific situations, *e.g.* when a strand-discrimination signal is buried within chromatin. In such difficult-to-repair situations, successful MMR may rely on the maintenance of the strand signals by MutSα. Since the leading strand seems to have less PCNA than the lagging strand (*Yu et al., 2014*), such situations could happen more frequently in the leading strand, in which MMR is less efficient than in the lagging strand (*Pavlov et al., 2003*; *Lujan et al., 2014*). An interesting possibility worth testing in the future would be that MutSα may take over PCNA from other factors such as DNA polymerases. Hijacking PCNA from polymerases may not be overly deleterious to DNA synthesis, because new PCNA loading at the 3'-terminus would quickly restore the PCNA-polymerase complex. Similar hijacking might occur on other PCNA interactors such as CAF-1 and FEN-1, both of which are reported to show complex interplay with MutSα (*Kadyrova et al., 2011*; *Schöpf et al., 2012*; *Kadyrova et al., 2015*; *Liu et al., 2015*). On the leading strand that has less PCNA, such competition could be important for effective MMR.

Much of our understanding of MMR has come from bacterial studies including *E. coli*, whose MMR utilizes adenine methylation as the strand discrimination signal. Although *E. coli* MMR seems to function with strand discontinuities occurring at the replication fork, GATC methylation plays a

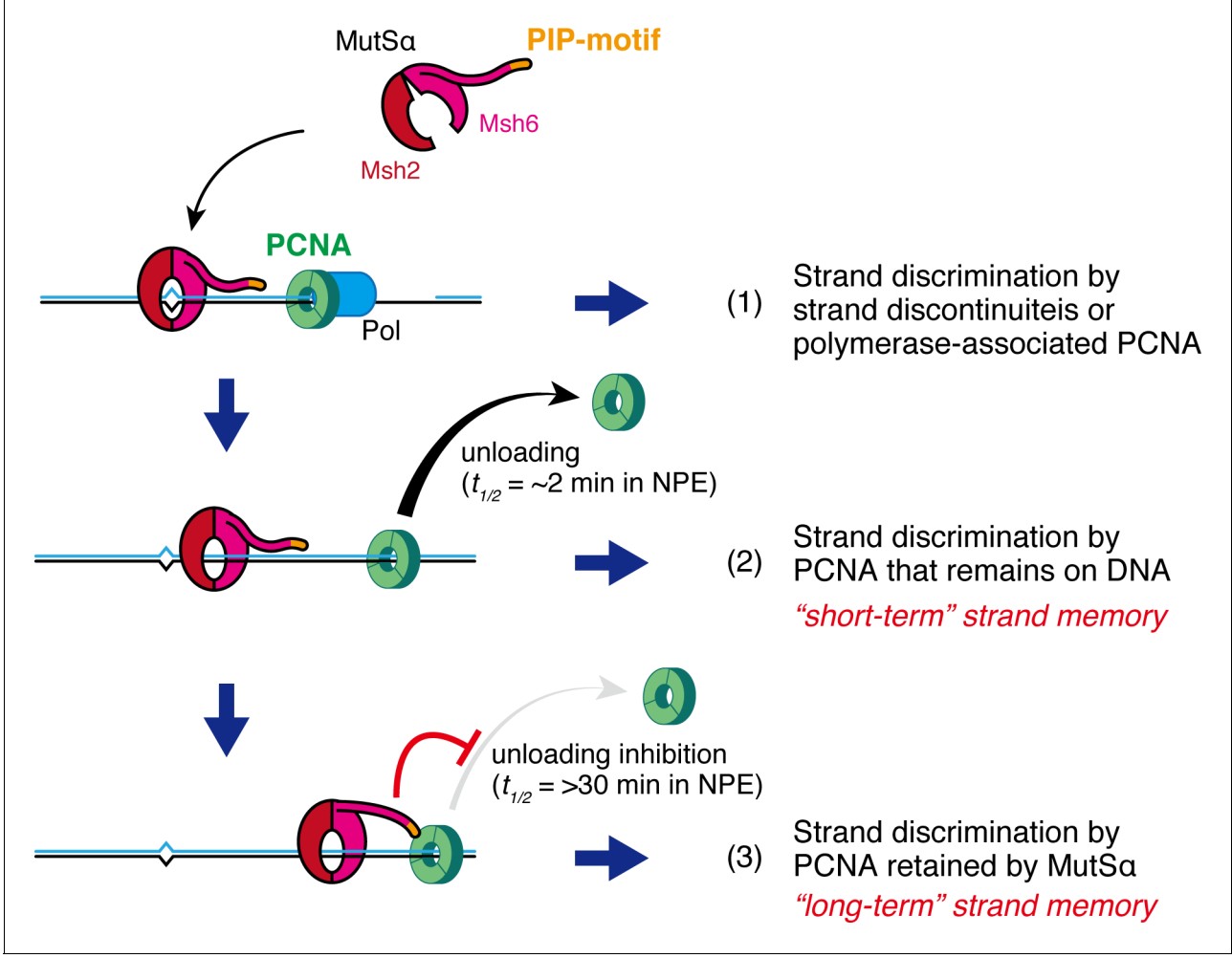

**Figure 10.** Two modes of strand memory maintain the MMR capability. Three mechanisms, including two 'strand-memory' mechanisms, may ensure eukaryotic MMR. (**1**) During ongoing DNA synthesis, strand discontinuities or polymerase-associated PCNA are directly used for strand discrimination. (**2**) After sealing of local strand discontinuities, PCNA that remains on DNA provides strand information until they are unloaded ($t_{1/2}$ = ~2 min in NPE; 'short-term' strand memory). (**3**) MutSα inhibits PCNA unloading to maintain the strand discrimination capability, largely through the PIP-motif on the NTR of Msh6 ($t_{1/2}$ = >30 min in NPE; 'long-term' strand memory). Black and blue lines represent the template and the newly synthesized DNA strands, respectively.

critical role in maintaining the temporal window permissive to MMR. Strand discontinuities are clearly strand discrimination signals in eukaryotic MMR in vitro (*Holmes et al., 1990*; *Thomas et al., 1991*). There is also concrete evidence that single-strand DNA termini such as 5'-ends of the Okazaki fragments or strand breaks generated by ribonucleotide excision repair contribute to MMR in vivo (*Pavlov et al., 2003*; *Nick McElhinny et al., 2010*; *Ghodgaonkar et al., 2013*; *Liberti et al., 2013*; *Lujan et al., 2013*; *2014*; *Liu et al., 2015*). We suggest that, in addition to such DNA termini, two modes of strand memory, somewhat resembling short-term and long-term memories in neuroscience, would operate as functional parallels of GATC methylation in *E. coli*. Analogous reactions could operate in bacteria and archaea lacking the methyl-dependent MMR (*Pillon et al., 2015*), and it would be interesting to study dynamics of the replication clamp in such organisms. Further investigation of the regulation of PCNA dynamics by MMR both in NPE and in vivo will lead to a more detailed understanding of complex interplay between the MMR system and the replication complex.

## Materials and methods

### Preparation of the MMR substrates

In vitro synthesis of mismatch-carrying plasmids was performed essentially as described previously (*Higashi et al., 2012*). A 5'-phosphorylated oligonucleotide (5'-CAGTAACATGGATC<u>X</u>CGAGA TGCAGTACGGTCACC-3'; for homo-duplex, X=T, and for A:C mismatch, X=C) was annealed to single-stranded DNA prepared by using the M13KO7 phage. To introduce a site-specific biotin modification, an additional oligonucleotide carrying a site-specific biotin-dT modification (5'-CGCCTTGATCGT[Biotin-dT]GGGAACCGGAGCTGAATGAAGC-3') was also added. Second-strand DNA synthesis was performed by using T7 DNA polymerase (New England Biolabs, MA, USA) and T4 DNA ligase (Nippongene, Tokyo, Japan). The mismatch-carrying DNA was treated with *Xho*I (Takara Bio, Kusatsu, Japan) to digest DNA whose mismatch base was edited by T7 DNA polymerase. The covalently closed products were purified with the Cesium Chloride density gradient ultracentrifugation. A site-specific gap was introduced as follows. In vitro synthesized pMM1 was doubly nicked with Nt.*Bbv*CI (for A-strand gap) or Nb.*Bbv*CI (for C-strand gap) at 37°C for 1 hr, purified, and incubated at 70°C for 20 min to dissociate the 15-nt fragment flanked by two *Bbv*CI sites from parental DNA. The DNA was then immediately loaded on a Microspin S-400HR Column (GE Healthcare, Little Chalfont, UK) to remove the 15-nt fragment. A site-specific nick was introduced by treating in vitro synthesized pMM2 with Nt.*Bbv*CI (for A-strand nick) or Nb.*Bbv*CI (for C-strand nick) at 37°C for 1 hr. Singly biotinylated plasmid DNA was bound to Sepharose beads as described previously (*Higashi et al., 2012*). All restriction enzymes were purchased from New England Biolabs unless otherwise indicated.

### In vitro PCNA-loading assay

Immobilized DNA (100 ng bound to 1 μl of Streptavidin-biotin-Sepharose beads) was incubated in 2 vol of mHBS (10 mM Hepes-NaOH pH 7.5, 0.05% Tween-20, 10 mM $MgCl_2$, 200 μM EDTA, 150 mM NaCl), containing 50 mM phosphocreatine, 25 μg/ml creatine phosphokinase, 2 mM ATP, 400 μM DTT, 145 ng/μl hPCNA, and 2.2 ng/μl hRFC at 32°C for 10 min. All reagents were purchased from Sigma Aldrich (MO, USA) unless otherwise indicated. The hPCNA-DNA complexes were then washed thrice with mHBS. 1 vol of the DNA beads was then incubated in 4 vol of ligation buffer (66 mM Tris-HCl pH 7.4, 6.6 mM $MgCl_2$, 10 mM DTT, 1 mM ATP) containing 25 units/μl T4 DNA Ligase at 16°C for 90 min. This was followed by washing thrice with mHBS, once with Egg lysis buffer (ELB; 10 mM Hepes-KOH pH 7.7, 2.5 mM $MgCl_2$, 50 mM KCl) containing 1 M KCl, and then once with ELB. To linearize DNA after PCNA loading, the hPCNA-DNA complexes were treated with either control buffer or buffer containing *Xmn*I and washed thrice with mHBS. To recover DNA from NPE, the hPCNA-DNA complexes were washed once with ELB containing 0.2% Triton X-100 and once with ELB.

### MMR and gap-filling assays in NPE

NPE was supplemented with 2 mM adenosine-triphosphate (ATP), 20 mM phosphocreatine, and 5 μg/ml creatine phosphokinase, pre-incubated at 22°C for 5 min, followed by addition of DNA substrates (20 ng/μl). A typical MMR reaction consists of 17.4 μl of NPE, 0.2 μl of 200 mM ATP, 0.4 μl of 1 M phosphocreatine, 0.02 μl of 5 mg/ml creatine phosphokinase, and 2 μl of DNA substrate (200 ng/μl in 10 mM Tris-HCl pH 7.4, 1 mM EDTA). To monitor DNA synthesis, reaction mixtures were supplemented with 2 μCi of $\alpha$-[$^{32}$P]-dCTP (PerkinElmer Japan, Tokyo, Japan) before addition of the MMR substrates. For the stepwise incubation experiments, an equal volume of fresh NPE was added to the reaction mixture. The samples were incubated at 22°C, and aliquots of samples (1.5 ~ 3 μl for most experiments) were stopped by addition of 100 μl of 1% SDS in 20 mM EDTA. DNA was purified by treatment with 50 μg/ml Proteinase K, extracted with Phenol/Chloroform, precipitated with ethanol, and dissolved in 10 mM Tris-HCl pH 7.4, 1 mM EDTA containing 10 μg/ml RNase at 2.5 ng/μl. To analyze the MMR efficiency (*%repair*), DNA synthesis (*%Gap-filling synthesis*) and the incorporation of radioactivity during MMR, 12 ng each of DNA was digested with following restriction enzymes: *Xmn*I, *Bam*HI-HF, *Xho*I, *Pac*I or *Drd*I. After agarose gel electrophoresis, DNA was stained with SYBR Gold nucleic acid stain (Life technologies, CA, USA), and scanned with Typhoon FLA9000 (GE Healthcare). Signal intensities were quantified using ImageQuant TL software (GE Healthcare).

To analyze the gap-filling efficiency (%*closed*), ethidium bromide containing agarose gel electrophoresis was performed to separate covalently closed DNA from gap/nick-carrying open-form DNA. The gel was stained with SYBR Gold nucleic acid stain, and then scanned by Typhoon FLA9000 using SYBR Gold specific setting (473 nm excitation laser and 510 nm Long pass filter). Calculation of *% closed* DNA is described in *Figure 1—figure supplement 5*. To moniter DNA synthesis, DNA was separated by agarose gel, stained with SYBR Gold, photographed, vacuum-dried on filter paper, contacted on a phosphor imaging plate (Fujifilm, Tokyo, Japan), and the $^{32}$P signals were scanned by Typhoon FLA9000. All agarose gel electrophoresis was performed with 0.8% agarose gel and 0.5× Tris-borate-EDTA buffer.

## Quantification of DNA-bound hPCNA and hRFC

DNA samples were linearized by *Xmn*I and separated using agarose gel electrophoresis, and quantified using known amounts of linear DNA as standards. Proteins recovered with the DNA-beads were separated by SDS-PAGE alongside with recombinant protein standards and probed with either hPCNA or hRfc2 antibodies. The number of hPCNA and hRFC on each plasmid (bound PCNA [RFC] / plasmid) were calculated by dividing the protein amounts from quantitative immunoblotting by bead-bound plasmid amounts calculated from linearized DNA. If 10 ng ($5.2 \times 10^{-15}$ mol) DNA is loaded, 1 ng hPCNA ($M_r = 8.61 \times 10^4$ as a trimer, $1.2 \times 10^{-14}$ mol) and hRFC ($M_r = 2.87 \times 10^5$ as a complex, $3.5 \times 10^{-15}$ mol) correspond to binding of approximately 2.3 and 0.7 molecules per plasmid, respectively.

## *Xenopus* egg extracts

Preparation of HSS and NPE was carried out as described previously (*Walter et al., 1998*; *Lebofsky et al., 2009*). In all experiments, plasmid DNA was incubated at 20 ng/μl concentration.

## Cloning and plasmids

The human Proliferating Cell Nuclear Antigen (*PCNA*) gene was amplified from pT7-PCNA (*Fukuda et al., 1995*) by PCR using primers 5′-GGAACATATGTTCGAGGCGCGCCTGGTCC-3′ and 5′-GGAAGGATCCCTAAGATCCTTCTTCATCCTCG-3′, digested with *Nde*I and *Bam*HI, and cloned into pET21a (Merck Millipore, MA, USA), resulting in pET21a-hPCNA. The *Xenopus laevis pcna* gene was amplified from a *Xenopus* egg cDNA library (a kind gift from Vladimir Joukov) by PCR using primers 5′-GGAACATATGTTTGAGGCTCGCTTGGTGC-3′ and 5′-GGAAGGATCCTTAA-GAAGCTTCTTCATCTTCAATCTTGG-3′, digested with *Nde*I and *Bam*HI, and cloned into pET21a, resulting in pET21a-xPCNA. The *Xenopus laevis msh2* gene was amplified from the *Xenopus* egg cDNA library by two-step PCR using primers 5′-AAAGCAGGCTCCACCATGGCTGTGCAGCCCAAA-GAGAAGTTG-3′ and 5′-ACAAGAAAGCTGGGTCTCCTGCAGGCAATCCCGTTTTGGTTCTGG-3′, and then primers 5′-GGGGACAAGTTTGTACAAAAAAGCAGGCTCCAC-3′ and 5′-GGGGACCAC TTTGTACAAGAAAGCTGGGTC-3′, and cloned into pDONR201 (Life technologies) using the Gateway BP reaction, resulting in pDONR-xMSH2. Cloning of *Xenopus laevis msh6* gene was performed as follows. A BLAST search using the human Msh6 protein sequence identified a *Xenopus laevis* EST clone, TC357542. Based on this sequence, the full-length *msh6* cDNA was amplified from *Xenopus* egg cDNA by two-step PCR using primers 5′-AAAGCAGGCTCCACTCATATGTCTAAG-CAAAAAACCCTCTTCAGCTTCTTCACC-3′ and 5′-ACAAGAAAGCTGGGTTGGTACCTTGGAG-CAACTTCAGCCGCTTGTGG-3′, and then primers 5′-GGGGACAAGTTTGTACAAAAAAGCAGGC TCCAC-3′ and 5′-GGGGACCACTTTGTACAAGAAAGCTGGGTC-3′, and cloned into pDONR201 using the Gateway BP reaction, resulting in pDONR-xMSH6. A FLAG tag was added to the C--terminus of xMsh6 by two-step PCR using primers 5′-AAACATATGTCTAAGCAAAAAACCCTC TTCAGC-3′ and 5′-GTCGTCCTTGTAGTCGGTACCGCCTTGGAGCAACTTCAGCCGC-3′, and then primers 5′-AAACATATGTCTAAGCAAAAAACCCTCTTCAGC-3′ and 5′-GGAACCTGCAGGTTACTTG TCATCGTCGTCCTTGTAGTCGG-3′, digested with *Nde*I and *Sse*8387I, and cloned into pDE1a, a derivative of the pDONR201 vector, resulting in pDE1a-xMSH6-FLAG. The xMsh6 PCNA-interacting peptide (PIP) motif mutant (xMsh6$^{PIP}$) was constructed by two-step PCR using pDE1a-xMSH6-FLAG as the template and primers 5′-AAAACCCTCTTCAGCGCGGCGACCAAGTCTCCCCCTGTTTCC-3′ and 5′-CCGTCCCCTCCTTGACTGTACTG-3′, and then primers 5′-AAACATATGTCTAAGGC-GAAAACCCTCTTCAGC-3′ and 5′-CCGTCCCCTCCTTGACTGTACTG-3′, digested with *Nde*I and

*Xho*I, and cloned into pDE1a-xMSH6-FLAG, resulting in pDE1a-xMSH6PIP-FLAG. The xMsh6 N-terminal deletion mutant (xMsh6$^{\Delta N}$) was constructed by PCR using pDE1a-xMSH6-FLAG as the template and primers 5'-GGAACATATGTCTGCCCCTGAGTCATTTGAATCACAGGC-3' and 5'-CCATGCGCCGACTTGTCTTGGC-3', digested with *Nde*I and *Xmn*I, and cloned into pDE1a-xMSH6-FLAG, resulting in pDE1a-xMSH6ΔN-FLAG. The *Xenopus laevis mlh1* gene was amplified from *Xenopus* egg cDNA by PCR using primers 5'-AAAGCAGGCTCCACCATGGCGGGAGTTATTCGGCGGCTGG-3' and 5'-ACAAGAAAGCTGGGTCTCCTGCAGGGCACCTTTCAAACACTTTATATAAGTCGGG-3', then primers 5'-GGGGACAAGTTTGTACAAAAAAGCAGGCTCCAC-3' and 5'-GGGGACCACTTTGTACAAGAAAGCTGGGTC-3', and cloned into pDONR201 using the Gateway BP reaction, resulting in pDONR-xMLH1. All sequences were confirmed after each PCR step.

Baculoviruses for expression of xMsh2, xMsh6$^{WT}$-FLAG, xMsh6$^{PIP}$-FLAG and xMsh6$^{\Delta N}$-FLAG were constructed by transferring x*MSH2*, x*MSH6*-FLAG, x*MSH6*PIP-FLAG and x*MSH6*ΔN-FLAG genes into BaculoDirect C-term Linear DNA (Life technologies) using the Gateway LR reaction.

Construction of pMM0, pMM1 and pMM2 was performed as follows: A synthetic linker prepared by annealing of 5'-phosphorylated oligonucleotides 5'-GAATTCAAGCTTAGTCTGTTCCATGTCATGCAAGATATCTTCAGTC-3', 5'-ACTGGGTGACCGTACTGCATCTCGAGATCCATGTTACTGCGTCAGT-3', 5'-CGCTAACAGTCACGAACTGCTGCAGGAATTCGTAC-3', 5'-GAATTCCTGCAGCAGTTCGTGACTGTTAGCGACTGACGCAGTAACA-3', 5'-TGGATCTCGAGATGCAGTACGGTCACCCAGTGACTGAAGATATCTT-3', and 5'-GCATGACATGGAACAGACTAAGCTTGAATTCAGCT-3', was inserted between the *Kpn*I and *Sac*I sites in pBluescriptII KS(-) (Stratagene, CA, USA), resulting in pMM0. A synthetic linker carrying two *Bbv*CI sites prepared by the annealing of 5'-phosphorylated oligonucleotides 5'-GCTCCTCAGCTTAATTAACCTCAGC-3' and 5'-AGCGCTGAGGTTAATTAAGCTGAGG-3' was inserted into the *Bsp*QI site in pMM0, resulting in pMM1. A synthetic linker carrying one *Bbv*CI site prepared by annealing of 5'-phosphorylated oligonucleotides 5'-GCTCCTCAGCATATGCCTCGC-3' and 5'-AGCGCGAGGCATATGCTGAGG-3' was inserted into the *Bsp*QI site in pMM0, resulting in pMM2.

## Protein expression and purification

Purification of hPCNA was carried out essentially as described previously (*Fukuda et al., 1995*), with minor modifications. Protein expression was induced in the *Escherichia coli* BL21(DE3) strain transformed with pET21a-hPCNA by addition of 0.1 mM Isopropyl β-D-1-thiogalactopyranoside (IPTG) to media for 20 hr at 20°C. hPCNA was purified using DEAE Sepharose Fast Flow, HiTrap Q-HP, Hi Load 16/60 Superdex 200 prep grade, and then MonoQ 5/50 GL (GE Healthcare) in this order. Recombinant xPCNA was expressed and purified using essentially the same method as hPCNA. Purification of the hRfc1-5 complex has been described previously (*Ohta et al., 2002*; *Shiomi et al., 2004*).

Purification of xMutSα was performed as follows: Recombinant proteins were expressed by co-infecting Sf9 insect cells with xMsh2 and xMsh6$^{WT}$-FLAG, xMsh6$^{PIP}$-FLAG or xMsh6$^{\Delta N}$-FLAG baculoviruses at 28°C in Sf-900 II SFM medium (Life Technologies) supplemented with 2% (v/v) fetal bovine serum (FBS). Cells were harvested, washed with Phosphate Buffered Saline (PBS) and frozen in liquid nitrogen. Cells were suspended in buffer S (25 mM Tris-HCl pH 7.4, 250 mM NaCl, 5 mM 2-mercaptoethanol, 1 mM EDTA, 2 mM phenylmethylsulfonyl fluoride [PMSF], 1 mM benzamidine and 1x cOmplete, EDTA-free [Roche life science, Penzberg, Germany]), and the lysates were centrifuged at 81,800 $\times g$ (30,000 rpm) for 30 min in a Beckman 50.2Ti rotor (Beckman Coulter, CA, USA). Cleared lysates were passed through a DEAE Sepharose Fast Flow column, and then a FLAG-M2 agarose column (Sigma Aldrich). The xMsh2-xMsh6-FLAG complexes were eluted from the FLAG-M2 resin using 50 μg/ml FLAG-peptide (Sigma Aldrich) in buffer S. The peak fractions were pooled and three-fold diluted with buffer A (25 mM Tris-HCl pH 7.4, 5% glycerol, 5 mM 2-mercaptoethanol, 1 mM EDTA and 0.1x cOmplete, EDTA-free), loaded on a MonoQ 5/50 GL column, and bound proteins were eluted with a 0–1 M NaCl linear gradient in buffer A. Peak fractions were loaded on a Hi Load 16/60 Superdex 200 prep grade column and eluted with buffer A containing 0.1 M NaCl. Fractions whose molecular weight correspond to ~2.5 × 10$^5$ (xMsh2: $M_r$ = 1.04 × 10$^5$, xMsh6-FLAG: $M_r$ = 1.50 × 10$^5$) were pooled, concentrated using Amicon Ultra (Merck Millipore), and frozen with liquid nitrogen as small aliquots.

## Immunological methods

Rabbit xMsh2 antiserum was raised against N-terminally His-tagged and C-terminally Strep-II-tagged full-length *Xenopus* Msh2 expressed in *E. coli*. Rabbit xMsh6 antiserum was raised against peptide NH$_2$-CNGSPEGLALHKRLKLLQ-COOH, corresponding to residues 1324–1340 of *Xenopus* Msh6. Rabbit xMlh1 antiserum was raised against N-terminally His-tagged, full-length *Xenopus* Mlh1 expressed in *E. coli*. Rabbit xPCNA antiserum was raised against full-length *Xenopus* PCNA expressed in *E. coli*. Rabbit xRfc3 antiserum was raised against peptide NH$_2$-CKKFMEDGLEAMMF-COOH, corresponding to residues 344–356 of *Xenopus* Rfc3. All antibodies except for xMlh1 and xPCNA were affinity purified using corresponding antigens. Rabbit xOrc2 antiserum was a kind gift from Johannes Walter (*Vashee et al., 2003*). hPCNA antibodies (MBL International Corporation, Nagoya, Japan, #MH-12-3) and hRfc2 antibodies (Abcam, Cambridge, UK, #ab3615) are commercially available. For immunoblotting, xMsh2, xMsh6, xMlh1, xOrc2, xPCNA and xRfc3 antibodies were used at 1:10,000 dilutions, hPCNA antibodies were used at a 1:2000 dilution, and hRfc2 antibodies was used at a 1:4000 dilution. HRP-conjugated Goat Rabbit IgG (H+L) antibodies (Jackson ImmunoResearch, PA, USA, #111-035-003), Goat Mouse IgG (H+L) antibodies (#115-035-146) or Donkey Goat IgG (H+L) antibodies (#705-035-003) were used at a 1:10,000 dilution as the secondary antibody for all immunoblottings except for quantitative immunoblottings for hPCNA, for which the same antibodies conjugated with Alexa Fluor 647 was used. To evaluate the depletion efficiency, 0.125 µl of NPE was loaded on SDS-PAGE, unless otherwise indicated.

For xMsh2/xMsh6 double depletion from NPE, 2 µg of Msh2 IgG, 0.5 µg of Msh6 IgG, and 2.9 µl of Msh6 serum were bound to 1 µl of the recombinant protein A-Sepharose ('PAS', GE Healthcare). For xMlh1 depletion, 3 vol of xMlh1 serum was bound to 1 vol of PAS. For xMsh2/xMsh6 and xMlh1 triple depletion, 2 µl of PAS bound 4 µg of Msh2 IgG, 1 µg of Msh6 IgG, and 5.8 µl of Msh6 serum was combined with 1 µl of PAS bound 3 µl of xMlh1 serum. For xRfc3 depletion from NPE, 5 µg of xRfc3 IgG was bound to 1 µl of PAS. To deplete NPE except for the depletion of xMsh2/xMsh6 and xMlh1 triple depletion, 0.2 vol of antibody-coupled PAS beads were incubated in 1 vol of extracts at 4°C for 1 hr, and the procedure was repeated once in all depletion experiments except for the depletion of xMsh2/xMsh6 or xRfc3 from NPE, for which the procedure was repeated twice. For xMsh2/xMsh6 and xMlh1 triple depletion from NPE, 0.3 vol of antibody-coupled PAS beads were incubated in 1 vol of extracts at 4°C for 1 hr, and the procedure was repeated twice. For xPCNA depletion from HSS, 3 vol of xPCNA serum was bound to 1 vol of PAS. A total of 0.2 vol of antibody-coupled PAS beads was incubated in 1 vol of HSS at 4°C for 1 hr, and the procedure was repeated thrice. In most cases, we depleted 40 ~ 60 µl of extracts for an experiment.

## Surface plasmon resonance

Affinity of the xMutSα complexes to DNA was analyzed using BIACORE 3000 (GE Healthcare) (*Lee and Alani, 2006*; *Shell et al., 2007*). A 42 bp biotin-conjugated oligo DNA 5'-GGGTGACCGTACTGCATCTCGAGATCCATGTTACTGCGTCAG-3'-[Biotin] was coupled to a Sensor Chip SA (GE Healthcare) until the SPR signal reached to ~100 response units. The sequence around the mismatch on pMM1$^{AC}$ was used to design the DNA substrate. A complementary oligonucleotide carrying a mismatch base 5'-CTGACGCAGTAACATGGATCCCGAGATGCAGTACGGTCACCC-3' was then flowed in on the Sensor tip to obtain double stranded DNA substrates carrying an A:C mismatch. Various concentrations of xMutSα$^{WT}$, xMutSα$^{PIP}$ and xMutSα$^{ΔN}$ complexes were flowed over the sensor chip for 3 min at 20 µl/min in running buffer (20 mM Hepes-NaOH pH 7.5, 100 mM NaCl, 1 mM DTT and 0.005% Tween-20) to analyze the association step, followed by running buffer containing no protein to monitor the dissociation step for 3 min. The signal from an empty flow cell was used for reference subtraction for all experiments. The chip surface was regenerated with 3 M NaCl for 1 min. Dissociation constants ($K_D$) were calculated using the BIAevaluation software v4.1.

## Accession numbers

The GenBank accession numbers for sequences of *Xenopus* *msh2*, *msh6* and *mlh1* mRNA reported in this paper are LC075519, LC075520, and LC075521, respectively.

## Acknowledgements

We thank Akira Yasui for critical reading of the manuscript, Vladimir Joukov for a *Xenopus* cDNA library, Johannes Walter and Shou Waga for antibodies, Kanae Taki for assistance in the cloning of the *xMSH6* gene, Chikako Tokumura for assistance in production of recombinant proteins, and Haruhiko Takisawa, Yumiko Kubota and Satoru Mimura for discussion and help with frog maintenance.

## Additional information

### Funding

| Funder | Grant reference number | Author |
| --- | --- | --- |
| Japan Society for the Promotion of Science | 13J01924 | Yoshitaka Kawasoe |
| Ministry of Education, Culture, Sports, Science, and Technology | 26114711 | Takuro Nakagawa |
| Ministry of Education, Culture, Sports, Science, and Technology | 23131507 | Tatsuro S Takahashi |
| Inamori Foundation | | Tatsuro S Takahashi |
| Uehara Memorial Foundation | | Tatsuro S Takahashi |
| Naito Foundation | | Tatsuro S Takahashi |
| Ministry of Education, Culture, Sports, Science, and Technology | 23657114 | Tatsuro S Takahashi |
| Ministry of Education, Culture, Sports, Science, and Technology | 25131712 | Tatsuro S Takahashi |
| Ministry of Education, Culture, Sports, Science, and Technology | 25711022 | Tatsuro S Takahashi |

The funders had no role in study design, data collection and interpretation, or the decision to submit the work for publication.

### Author contributions

YK, Conception and design, Acquisition of data, Analysis and interpretation of data, Drafting or revising the article; TT, Drafting or revising the article, Contributed unpublished essential data or reagents; TN, HM, Analysis and interpretation of data, Drafting or revising the article; TST, Conception and design, Analysis and interpretation of data, Drafting or revising the article

### Author ORCIDs

Yoshitaka Kawasoe, http://orcid.org/0000-0003-0925-2004
Toshiki Tsurimoto, http://orcid.org/0000-0001-7597-2216
Takuro Nakagawa, http://orcid.org/0000-0003-3455-8224
Tatsuro S Takahashi, http://orcid.org/0000-0002-1947-7680

### Ethics

Animal experimentation: Animal handling and maintenance were performed according to protocols approved by the Osaka University Institutional Animal Care and Use Committee (Permit Number: DOURI-27-02-0).

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
