## [Decision Letter]

Thank you for submitting your article "MutSα maintains the mismatch repair capability by inhibiting PCNA unloading" for consideration by *eLife*. Your article has been favorably evaluated by John Kuriyan (Senior editor) and three reviewers, one of whom is a member of our Board of Reviewing Editors.

The reviewers have discussed the reviews with one another and the Reviewing Editor has drafted this decision to help you prepare a revised submission.

The manuscript describes a series of experiments that analyze the strand-directed mismatch repair reaction in *Xenopus* egg extracts. This is an interesting and important work that advances our understanding of eukaryotic mismatch repair. However, the following important technical and conceptual issues need to be satisfactorily addressed before the paper can be considered for publication.

Major issues:

1) PIP-deficient MutS protein is proficient in MMR. Based on the authors' assumption, MutSα without the PIP box-containing N-termini should be defective in capturing strand discrimination signal from PCNA. Unfortunately, the mutant MutSα is as active as WT MutSα in repair of heteroduplexes bound by PCNA.

2) PCNA polarity and nick-mismatch orientation. Previous studies using a circular heteroduplex have established that regardless of nick orientation (i.e., 5' or 3') to the mismatch, MMR always occurs along the shorter distance between these two sites. If MutS protein interacts with PCNA through the front face of PCNA, MutSα and other MMR proteins recruited by MutSα to A-gap and C-gap substrates should be lined up in different polarities/orientations, which would suggest that repair in one substrate occurs along the shorter spin and the other in longer spin. These assumptions are not consistent with what is known.

3) Interaction between PCNA and MutSα. Although MutS protein has been shown to be able to slide along the DNA, increasing evidence (both in vivo and in vitro) suggests that MutS protein does not travel once it binds to a mismatch. The authors suggest that the stand signal carried by PCNA is transmitted to MutSα when the latter interacts with PCNA through its PIP box. Given that the mismatch is ~300 bp away from the gap site, where PCNA is located, it is impossible for MutS protein to reach PCNA. How could this interaction occur?

4) The authors showed that the presence of PCNA in closed circular heteroduplex is essential for strand-specific repair, and that the DNA bound PCNA could be unloaded within ~5 min. What is the mechanism by which PCNA is released from a closed circular DNA molecule?

Other specific issues:

1) The deletion of the PIP motif of MSH6, which the authors have identified as a major determinant involved in the inhibition of PCNA unloading by MutSalpha, decreased the in vivo mismatch repair activity of yeast MutSalpha in MSH3-deficient cells by ~5-20% (Table 1 in Clark et al. (2000) JBC, 275, 36498). Thus, it appears that the inhibition of PCNA unloading by MutSalpha is not essential for MMR. Some discussion of this fact would be helpful.

2) In the Abstract, the authors state that "Our data identify PCNA as the signal that enables strand discrimination after the completion of DNA synthesis…". I think they should write something like "Our data identify loaded PCNA as the signal that enables strand discrimination after the strand discontinuity is removed…".

3) The description of the MMR and gap-filling assays and immunological methods in Materials and methods and figure legends is too vague and lacks many critical experimental details including the final volumes and ionic strength of the reaction mixtures and the amounts of the extracts present in the reaction mixtures. The lack of this information complicates understanding the results. Please provide the relevant information.

4) It is not clear why the authors do not see the nick-directed MMR in the NPE. In fact, the data shown in panel B of Figure 4—figure supplement 1 suggest that MMR is directed, though not very efficiently, to the covalently closed strand. This is somewhat confusing. Did the authors study whether the rates of ligation of the nicked and gapped DNA substrates differ from each other? Some experimental data would be helpful to explain the failure to see the nick-directed MMR.

5) Figure 3: it appears that the fractions of the nicked form in the ligated DNA samples analyzed in lanes 1-4 of C panel are significantly higher than those in panel B (lanes 5-8). It appears that at least 35% of DNA in lane 4 of panel C is nicked. Please provide the percentages of the closed form present in the samples analyzed in the panel C lanes. Also indicate the percentage of the closed form for each of the substrates used in the reactions in D.

6) Inspection of the repair data in panel B of Figure 4—figure supplement 1 suggests that in several cases the band intensities do not correspond well to the indicated repair values. Please provide the averages and S.D. for the experiments in this panel.

In some control reactions (panel B of Figure 4—figure supplement 1) the level of C-to-T repair on the continuous strand is about 25% and about 50% higher than that on the nicked strand. How do the authors explain this result? Is this the strand that was synthesized in vitro?

7) MMR reactions observed in the NPE display poor strand discrimination. It is evident that the discontinuous strand is repaired only 5-10 times better than the continuous strand. Such weak strand discrimination would cause many mutations in vivo, but in vivo data indicate that the strand discrimination is very efficient. Please discuss potential reasons of the limited strand discrimination in the analyzed reactions.

8) Did the authors verify in Figure 6 that MutS depletion did not somehow affect the rate of closed circular plasmid accumulation?

9) In Figure 1, it would be helpful to indicate which samples were digested with XhoI, which with BamHI. Also, the difference between panels 1D and E is not apparent.

10) There is a major discrepancy in the kinetics of% closed plasmid accumulation between Figure 1—figure supplement 4 (slow) and Figure 1—figure supplement 3 (fast) (and Figure 2, also fast). Which is correct? It seems that the slow kinetics of closed circular formation (black graph in 1F) is therefore not representative.

11) "Unlike homoduplex plasmids, the gap-filling DNA synthesis did not covalently close the mismatch-carrying plasmid, suggesting that new strand breaks are introduced on the DNA." This sentence is unclear. Do the authors mean that the mismatch plasmid was not fully converted to supercoiled, based on the ~20% fraction of gapped/nicked molecules remaining in Figure 1—figure supplement 3? Overall, this paragraph is difficult to follow.

12) "Although it has not been clear whether the MMR system can correct replication errors after the completion of local DNA synthesis except the ribonucleotide-directed mechanism, since PCNA remains on DNA after the completion of DNA synthesis until unloaded by the PCNA unloader complex, DNA-bound PCNA could mediate the coupling of MMR with DNA replication by generating a post-replicative temporal window permissive to MMR." This sentence is too long and poorly constructed.

13) "In the absence of MutSα/β, the strand information was retained for 1-2 min after disappearance of strand discontinuities (Figure 2)" This is an odd description of the data in Figure 2. It would be better to say: "In the absence of MutSα/β, the strand information was gradually lost after the disappearance of strand discontinuities…".

---

## [Author Response]

*Major issues:*

1) PIP-deficient MutS protein is proficient in MMR. Based on the authors' assumption, MutSα without the PIP box-containing N-termini should be defective in capturing strand discrimination signal from PCNA. Unfortunately, the mutant MutSα is as active as WT MutSα in repair of heteroduplexes bound by PCNA.

As the reviewers pointed out, we did not see strong reduction of the MMR activity by the deletion of the PIP-containing N-terminus of MutSα (Figure 8—figure supplement 1). We think that there are two important implications from this fact.

In this paper, we have shown that (1) DNA-bound PCNA can induce strand specific MMR in NPE, and that (2) the PIP-containing N-terminus of MutSα inhibits unloading of PCNA to maintain the strand information derived from DNA-bound PCNA. On the one hand, the PIP motif in MutSα should be in principle dispensable for MMR. Previous genetic studies in yeast have shown that the PIP motif is important but not essential for MMR. Three previous studies (Kleczkowska et al., *Genes Dev*, 2001, Iyer et al., *J Biol Chem*, 2008, and Bowen et al., *Proc Nat Acad Sci*, 2013) have already shown that PIP-deficient MutSα can promote MMR either in human cell extracts or in a reconstituted system. Although Kleczkowska et al. have seen a partial reduction in the MMR efficiency with PIP deletion, other two studies agree that PIP-deficient MutSα is essentially as active as the wild-type protein for in vitro MMR that uses plasmid substrates. A series of seminal studies from the Modrich lab has shown that it is MutLα that initiates strand removal during MMR. This point is related to Major issue 2, and discussed there in detail (Discussion, third paragraph). Both in vivo data and in vitro analyses indicate that the PIP motif in MutSα is not essential for activation of MutLα.

On the other hand, since inhibition of PCNA unloading would indirectly assist strand discrimination through the retention of the strand signal, PIP-defective MutSα should have somewhat reduced activity in repair of mismatches. As mentioned already, although not essential, the PIP motif has an important role in MMR, since its deletion shows substantial increment of spontaneous mutation rates in yeast cells. The discrepancy between the in vivo observation and in vitro experiments suggest that (1) the PIP-dependent inhibition of PCNA unloading is important under specific situations that occur in vivo, and (2) plasmid-based in vitro MMR systems may not recapitulate such situations. Interestingly, once stepwise incubation, which mimics transient arrest of the MMR reaction, was involved, the PIP motif became important for the MMR efficiency in an in vitro assay (Figure 9). Therefore, it would be reasonable to speculate that the PIP motif becomes important for MMR under a certain “difficult-to-repair” situation, e.g. when PCNA is buried in chromatin. We now included relevant discussion in the revised text (Discussion, seventh paragraph). We feel that addition of above arguments made the paper significantly richer, and appreciate the reviewers’ suggestion.

2) PCNA polarity and nick-mismatch orientation. Previous studies using a circular heteroduplex have established that regardless of nick orientation (i.e., 5' or 3') to the mismatch, MMR always occurs along the shorter distance between these two sites. If MutS protein interacts with PCNA through the front face of PCNA, MutSα and other MMR proteins recruited by MutSα to A-gap and C-gap substrates should be lined up in different polarities/orientations, which would suggest that repair in one substrate occurs along the shorter spin and the other in longer spin. These assumptions are not consistent with what is known.

In the original manuscript, we have not traced DNA synthesis during MMR, and therefore it has not been clear whether NPE supports bidirectional strand excision. In response to the reviewers’ criticism, we have carried out this experiment. The result showed that, as widely acknowledged, repair DNA synthesis occurs preferentially along the shorter path between a gap and a mismatch, regardless of relative orientation between them (new Figure 1—figure supplement 2). The data confirmed that NPE promotes truly bidirectional MMR with the capability of both 5’ to 3’ and 3’ to 5’ strand excision (Results, first paragraph).

As mentioned in Major issue 1, biochemical reconstitution studies from the Modrich lab (Kadyrov et al., *Cell*, 2006, Kadyrov et al., *J Biol Chem*, 2007, Pluciennik et al., *Proc Natl Acad Sci*, 2010, Pluciennik et al., *Proc Natl Acad Sci*, 2013) have shown that MutLα is the key enzyme to initiate and promote strand excision. Since both gap-directed and PCNA-directed MMR was dependent on the MutL complexes, we assume that, as widely accepted, bidirectional strand excision seen in NPE is initiated by MutLα. As the reviewers pointed out, either or both of MutSα and MutLα should be lined up toward a specific face of PCNA, and it is interesting to understand how this complex degrades the DNA strand in either direction depending on the relative orientation toward a mismatch. Although clarification of this mechanism will require extensive future work such as single-molecule tracing of the MMR machinery and will be beyond the scope of this paper, discussion on this point would be important for interpretation of the data presented in this study. We now included relevant discussion in the revised manuscript (Discussion, third paragraph).

In addition, previous in vitro studies have shown that the 5’ to 3’ and 3’ to 5’ strand excision reactions employ a slightly different set of proteins, suggesting mechanistic difference between two modes of strand excision; the former does not require MutLα, while the latter requires it. We observed that both 5’-gap-directed and 3’-gap-directed MMR reactions were dependent on the MutL complexes in NPE (Figure 1). We thought that this finding is also related to the issue discussed above, and we included discussion on this point as well (Discussion, second paragraph).

3) Interaction between PCNA and MutSα. Although MutS protein has been shown to be able to slide along the DNA, increasing evidence (both in vivo and in vitro) suggests that MutS protein does not travel once it binds to a mismatch. The authors suggest that the stand signal carried by PCNA is transmitted to MutSα when the latter interacts with PCNA through its PIP box. Given that the mismatch is ~300 bp away from the gap site, where PCNA is located, it is impossible for MutS protein to reach PCNA. How could this interaction occur?

Since the PIP motif is not essential for MMR, it will not be essential for transmission of the strand information from PCNA to the MMR system. However, we suggested in this paper that MutSα may capture DNA-bound PCNA possibly from a remote place through the PIP-motif. In this case, the mismatch-bound MutSα (and likely MutLα also) and PCNA that is present at a different place would eventually need to interact with each other to initiate MMR. As the reviewers have commented, there is evidence that the MutS/MutL complexes binding to a mismatch do not slide away from the mismatch, and multiple MutL complexes may be responsible for communication between mismatch-bound MMR complexes and nearby PCNA molecules. We apologize for the lack of mention on this point in the original manuscript. We now included discussion on potential mechanisms of how MutSα (and MutLα) and a PCNA molecule that is captured by MutSα can communicate with each other to initiate MMR (Discussion, fifth paragraph). We appreciate the reviewers’ helpful suggestion on this point.

4) The authors showed that the presence of PCNA in closed circular heteroduplex is essential for strand-specific repair, and that the DNA bound PCNA could be unloaded within ~5 min. What is the mechanism by which PCNA is released from a closed circular DNA molecule?

Since we used a closed circular DNA substrate for our assay, PCNA unloading should involve the opening of the PCNA ring, a reaction that is catalyzed by the PCNA loader/unloader complex(es). Indeed, we showed that depletion of Rfc3, a common subunit of all four RFC-like complexes, significantly slows down the unloading reaction in NPE. It has been known that canonical RFC can catalyze both loading and unloading of PCNA. In addition, the Elg1-containing RFC like complex is largely responsible for PCNA unloading from post-replicative DNA during S phase. Therefore, we speculate that either or both of them would be involved in the unloading of PCNA from circular DNA substrates in NPE. Unavailability of specific antibodies against *Xenopus* Rfc1 or Elg1 prevented us to distinguish which one is primarily responsible for PCNA unloading. However, the identity of the relevant unloader does not affect our conclusion that MutSα prevents the unloading of PCNA. In the original manuscript, we just briefly mentioned that PCNA unloading is dependent on an unloading complex(es) containing Rfc3. In response to the reviewers’ point, we now included discussion on possible factors and mechanism of PCNA unloading and speculation on the mechanism of the inhibition of PCNA unloading by MutSα (Discussion, fourth paragraph), which we feel significantly improved the paper.

*Other specific issues:*

1) The deletion of the PIP motif of MSH6, which the authors have identified as a major determinant involved in the inhibition of PCNA unloading by MutSalpha, decreased the in vivo mismatch repair activity of yeast MutSalpha in MSH3-deficient cells by ~5-20% (Table 1 in Clark et al. (2000) JBC, 275, 36498). Thus, it appears that the inhibition of PCNA unloading by MutSalpha is not essential for MMR. Some discussion of this fact would be helpful.

We thank the reviewers for the helpful suggestion. As discussed in Major issue 1, we predict that the PIP motif may be needed for MMR under a certain “difficult-to-repair” situation, e.g. when PCNA is buried in chromatin. We included relevant discussion in the revised manuscript (Discussion, seventh paragraph).

*2) In the Abstract, the authors state that "Our data identify PCNA as the signal that enables strand discrimination after the completion of DNA synthesis…". I think they should write something like "Our data identify loaded PCNA as the signal that enables strand discrimination after the strand discontinuity is removed…".*

We appreciate the reviewers’ correction. The sentence in the Abstract was changed as follows: “Our data identify DNA-bound PCNA as the signal that enables strand discrimination after disappearance of strand discontinuities, and uncover a novel role of MutSα in the retention of the post-replicative MMR capability.”

3) The description of the MMR and gap-filling assays and immunological methods in Materials and methods and figure legends is too vague and lacks many critical experimental details including the final volumes and ionic strength of the reaction mixtures and the amounts of the extracts present in the reaction mixtures. The lack of this information complicates understanding the results. Please provide the relevant information.

Unlike other in vitro systems that use cell extracts, *Xenopus* egg extracts are prepared essentially devoid of additional buffers, and usually used as is, without addition of any buffers or salts. However, we realized that, in the original version of Materials and methods, this was not very clear for readers, and also we didn’t mention the volumes of the reactions. We now included relevant information in Materials and methods and figure legends. Also we added more information regarding the immunodepletion procedure to describe better what we have carried out. We apologize for the lack of the relevant information in the original manuscript.

4) It is not clear why the authors do not see the nick-directed MMR in the NPE. In fact, the data shown in panel B of Figure 4—figure supplement 1 suggest that MMR is directed, though not very efficiently, to the covalently closed strand. This is somewhat confusing. Did the authors study whether the rates of ligation of the nicked and gapped DNA substrates differ from each other? Some experimental data would be helpful to explain the failure to see the nick-directed MMR.

A nick can induce some strand specific MMR but not as efficiently as a gap, and we originally stated “Unlike other systems, a nick did not efficiently induce MMR, possibly because nick ligation is very quick in NPE”. Upon the reviewers’ suggestion, we now showed a direct comparison between nick- and gap-directed MMR (new Figure 1—figure supplement 4) and performed a follow-up experiment, in which we found that ligation of a nick is only partially dependent on PCNA (new Figure 3—figure supplement 1, panels E and F). Since sealing of a gap is completely dependent on PCNA, this new data provides an explanation as to why nicks do not induce strand-specific MMR as efficiently as gaps. We now included these new data, and modified relevant descriptions accordingly (subsection “The MutS complexes inhibit Rfc3-dependent PCNA unloading”, first paragraph).

5) Figure 3: it appears that the fractions of the nicked form in the ligated DNA samples analyzed in lanes 1-4 of C panel are significantly higher than those in panel B (lanes 5-8). It appears that at least 35% of DNA in lane 4 of panel C is nicked. Please provide the percentages of the closed form present in the samples analyzed in the panel C lanes. Also indicate the percentage of the closed form for each of the substrates used in the reactions in D.

In the experiment shown in Figure 3, we split the samples shown in (B) into two portions. One portion (C) was incubated in the presence or absence of *Xmn*I for 60 min to see if PCNA slides off from DNA by linearization of the plasmid, and the other portion (D) was immediately incubated in NPE to see if MMR occurs. The nicked form plasmids seen in (C), lanes 1-4, was therefore accumulated during incubation in a buffer for 60 min. The percentages of the closed form for substrates used in (D) are exactly those shown in (B). We apologize for this confusion, and modified the legend for Figure 3 to make this point clear. The percentages of the closed circular molecules for lanes 1-4 are 63, 60, 62, and 59%, respectively. Since we considered that the percentages of the closed form in (C) are not relevant for interpretation of the results, we did not put these values in panel (C) in the revised manuscript.

*6) Inspection of the repair data in panel B of Figure 4—figure supplement 1 suggests that in several cases the band intensities do not correspond well to the indicated repair values. Please provide the averages and S.D. for the experiments in this panel.*

We repeated the quantification of the data in panel (B) of Figure 4—figure supplement 1, and confirmed that the values presented in the figure are correct. We are ready to provide the original tiff image and/or the raw quantification data upon reviewers’ request. To provide the averages and standard deviation for the experiment, we now also included a graph of the quantification data from three independent experiments with standard deviation (Figure 4—figure supplement 1, panel C).

In some control reactions (panel B of Figure 4—figure supplement 1) the level of C-to-T repair on the continuous strand is about 25% and about 50% higher than that on the nicked strand. How do the authors explain this result? Is this the strand that was synthesized in vitro?

We have made the substrates by in vitro synthesis of the second strand using single-stranded phagemid DNA as the templates (corresponding to the “A”-strand). Therefore, the observed C-to-T repair occurred on the strand synthesized in vitro. It is not clear why the level of repair on the continuous strand was higher than that on the nicked strand. However, as shown in the new Figure 4—figure supplement 1, panel C, this tendency was not statistically significant. Perhaps more mechanical or chemical DNA damages have occurred on the substrate in this particular experiment for some reasons.

Since the substrates were synthesized in vitro, it is likely that the bases in free nucleotides used as substrates for in vitro DNA synthesis and/or on single-stranded DNA templates contain small amount of spontaneous damages (e.g. oxidization of bases). Such damages can elicit base-excision repair (BER), which in turn trigger MMR by providing strand breaks (Repmann et al., *J Biol Chem*, 2015). We predict that such BER may be part of the reason why we see some background repair of mismatches. In some substrates, the background repair showed slight strand bias as pointed out by the reviewers, and this may be because a specific strand contained more damages, although we don’t have evidence for this point. We now included short description of DNA preparation, and relevant arguments in the revised text (Results, first paragraph).

7) MMR reactions observed in the NPE display poor strand discrimination. It is evident that the discontinuous strand is repaired only 5-10 times better than the continuous strand. Such weak strand discrimination would cause many mutations in vivo, but in vivo data indicate that the strand discrimination is very efficient. Please discuss potential reasons of the limited strand discrimination in the analyzed reactions.

We agree that the strand bias of MMR seen in NPE was not strong enough to explain the in vivo MMR efficiency. Since gap-directed repair of mismatch is reasonably efficient as an in vitro reaction and sometimes reaches to 80% (e.g. Figure 1), the major reason for this somewhat modest strand bias is relatively high non-specific repair of mismatches. As discussed in the response to specific point 6, we speculate that such non-specific repair could be due to background BER or physical damages on the DNA substrate. As stated there, we included the relevant argument in the revised text (Results, first paragraph).

8) Did the authors verify in Figure 6 that MutS depletion did not somehow affect the rate of closed circular plasmid accumulation?

Although we did not examine whether depletion of MutS or MutL affects the rate of closed plasmid accumulation in this particular experiment shown in Figure 6, Figure 7 shows that these depletions do not significantly affect either the rate of gap filling or the level of the closed circular molecules. In addition, we have repeated the experiment shown in Figure 6 to test accumulation of closed circular molecules in MutS- or MutL-depleted NPE (new Figure 6—figure supplement 2). The results show that neither of depletion conditions largely affects the rate of closed circular plasmid accumulation, and importantly, nicks were not accumulated in MutL-depleted NPE in which we observed longer retention of the MMR capability. Therefore, it is unlikely that difference in the accumulation of closed circular molecules has affected the MMR capability after incubation in MutS- or MutL-depleted NPE.

9) In Figure 1, it would be helpful to indicate which samples were digested with XhoI, which with BamHI. Also, the difference between panels 1D and E is not apparent.

We thank the reviewers for their helpful suggestion. We now included the information of the restriction enzyme in the legend of Figure 1. For panels 1D and 1E, we changed the order of the panels in a way such that MMR in MutS-depleted NPE (former 1D) immediately follows the western blot (1B), and MMR in MutL-depleted NPE (1E) does so as well.

10) There is a major discrepancy in the kinetics of% closed plasmid accumulation between Figure 1—figure supplement 4 (slow) and Figure 1—figure supplement 3 (fast) (and Figure 2, also fast). Which is correct? It seems that the slow kinetics of closed circular formation (black graph in 1F) is therefore not representative.

Both data are actually correct. Figure 1—figure supplement 3 shows the kinetics of sealing of a gap in the absence of a mismatch, and Figure 1—figure supplement 4 is the same experiment but in the presence of a mismatch. We agree that the difference was not explicit in the original manuscript, and apologize for this confusion. We now revised the description of the results to make this point clear (Results, second paragraph). In Figure 2, the plasmid substrate was incubated in MutS-depleted NPE. Therefore, although the substrate contained a mismatch, it behaved as if there was no mismatch present on the DNA.

11) "Unlike homoduplex plasmids, the gap-filling DNA synthesis did not covalently close the mismatch-carrying plasmid, suggesting that new strand breaks are introduced on the DNA." This sentence is unclear. Do the authors mean that the mismatch plasmid was not fully converted to supercoiled, based on the ~20% fraction of gapped/nicked molecules remaining in Figure 1—figure supplement 3? Overall, this paragraph is difficult to follow.

This point is related to specific point 10. We apologize for the ambiguous description of the experiments, and revised the text in order to make it clear that we have carried out two independent experiments with a substrate carrying no mismatch and that carrying a mismatch (Results, second paragraph).

12) "Although it has not been clear whether the MMR system can correct replication errors after the completion of local DNA synthesis except the ribonucleotide-directed mechanism, since PCNA remains on DNA after the completion of DNA synthesis until unloaded by the PCNA unloader complex, DNA-bound PCNA could mediate the coupling of MMR with DNA replication by generating a post-replicative temporal window permissive to MMR." This sentence is too long and poorly constructed.

We thank the reviewers’ criticism. To state only essential points, the sentence was simplified as follows. “Since PCNA transiently remains on DNA after the completion of DNA synthesis, DNA-bound PCNA could mediate the coupling of MMR with DNA replication.”

*13) "In the absence of MutSα/β, the strand information was retained for 1-2 min after disappearance of strand discontinuities (Figure 2)" This is an odd description of the data in Figure 2. It would be better to say: "In the absence of MutSα/β, the strand information was gradually lost after the disappearance of strand discontinuities…".*

The reviewers are correct. We changed the expression accordingly (subsection “The MutS complexes inhibit Rfc3-dependent PCNA unloading”, first paragraph).